

# Sedimentary architecture and landforms of the Late Saalian (MIS 6) ice sheet margin, offshore the Netherlands

Víctor Cartelle[1], Natasha L.M. Barlow[1], David M. Hodgson[1], Freek S. Busschers[2], Kim M. Cohen[3], Bart M.L. Meijninger[2], Wessel P. van Kesteren[4]

[1]School of Earth and Environment, University of Leeds, UK
[2]TNO, Geological Survey of the Netherlands, Utrecht, The Netherlands
3Faculty of Geosciences, Utrecht University, Utrecht, The Netherlands
[4]Fugro, Nootdorp, The Netherlands

*Correspondence to*: Víctor Cartelle (v.cartellealvarez@leeds.ac.uk)

**Abstract.** Reconstructing the growth and decay of palaeo-ice sheets is critical to understanding the relationships between global climate and sea-level change, and to testing numerical ice sheet models. In this study, we integrate recently acquired high-resolution 2D-seismic reflection and borehole datasets from two windfarm sites offshore the Netherlands to investigate the sedimentary, geomorphological and glaciotectonic records left by the Saalian Drenthe substage glaciation, when Scandinavian land ice reached its southernmost extent in the southern North Sea (ca. 160 ka, Marine Isotope Stage 6). A complex assemblage of glaciogenic sediments and glaciotectonic structures are buried in the shallow subsurface. The northern windfarm site revealed a set of NE-SW oriented subglacial meltwater channels filled with till and glaciofluvial sediments and an E-W trending composite ridge with local evidence of intense glaciotectonic deformation that denotes the maximum limit reached by the ice. Based on the identified glacial geomorphology, we refine the mapping of the maximum ice-sheet extent offshore the Netherlands, revealing that the ice margin morphology is more complex than previously envisaged, displaying a lobate shape. Ice retreat left an unusual paraglacial landscape characterised by the progressive infilling of topographic depressions carved during the ice advance and a diffuse drainage network of outwash channels. The net direction of outwash was to the west and southwest into a nearby glacial basin. Antecedent topography influenced subglacial bed conditions, and their impact on ice dynamics during the glaciation and deglaciation stages. We demonstrate the utility of offshore windfarm data in refining palaeo ice margin limits, and the record of processes interactions preserved in buried landscapes to help inform longer-term drivers of change at low relief ice margins.

Earth **Surface**
**Dynamics**
Discussions
EGU

## 1 Introduction

The Greenland and Antarctic Ice Sheets have experienced net mass loss since the end of the Last Glacial (Alley et al., 2010; Anderson et al., 2002; Prothro et al., 2020), and are continuing to shrink at increasing rates due to anthropogenically-driven climatic change and oceanic warming (Hanna et al., 2020; Shepherd et al., 2020, 2019). Therefore, it is important to better

constrain the driving mechanisms of modern and future ice-sheet changes, and the potential for global and regional sea-level rise. Observational data of modern ice sheet mass-balance cannot capture the temporal and spatial patterns of large-scale ice-sheet decay. Therefore, there is a need to develop reconstructions of past ice-sheet advance and retreat to understand changes on longer timescales to improve assessment of the duration and magnitude of ice-sheet instabilities; to understand lags in the Earth-system response; and the potential for changes in relative sea level (RSL) (Carlson and Clark, 2012). Geological records

can be used to inform and calibrate models of ongoing ice-sheet retreat and sea-level rise, which may provide explanations for modern-day observations and be used to estimate potential future changes (Clark and Huybers, 2009; Gilford et al., 2020; Harrison et al., 2015; Kleman et al., 2007; Stokes et al., 2015).

To further understand ice sheet responses to differing climate states, we focus on ice-sheet advance and retreat during the penultimate glacial cycle (late Marine Isotope Stage [MIS] 6) in NW Europe. This period directly preceded the Last

Interglacial (LIG; ca. 129-116 ka, MIS 5e) during which global temperatures were ~1°C warmer than pre-industrial values (Otto-Bliesner et al., 2013), and global mean sea level was likely 6 to 9 m above present (Dutton et al., 2015). Together with climatic forcing and associated ice sheet responses over the MIS 6/5 glacial termination, the distribution of global ice sheets during MIS 6 is critical to understanding the nature of the LIG highstand regionally and globally. This is not trivial as much evidence of MIS 6 glaciation and deglaciations was removed or heavily overprinted by glaciations during the Last Glacial.

Thus, the uncertainty in knowing the MIS 6 ice-sheet limits and the timing of deglaciation (Batchelor et al., 2019), represent a weakness in glacial isostatic adjustment modelling, which is needed to understand rates and magnitudes of global and regional LIG RSL change (Barlow et al., 2018; Dendy et al., 2017; Rohling et al., 2017).

Broadly speaking, the MIS 6 North European ice sheet reached a greater eastward and southward extent compared to the Last Glacial Maximum (LGM, MIS 2), where it had terrestrial limits. Its marine-limited western extent was smaller than, or similar

to, that of the LGM (Batchelor et al., 2019; Ehlers and Gibbard, 2004). Over Germany and The Netherlands, the MIS 6

Earth **Surface**
**Dynamics**
Discussions
EGU
maximum ice advance was more extensive than the LGM, but even here reconstructions of ice limits are largely derived from spatially disparate field evidence that usually relate to small sectors of the ice sheet (e.g. Busschers et al., 2008; Gibbard and Clark, 2011; Van den Berg and Beets, 1987). Offshore in the southern North Sea, the current geological constraints on the position of the Saalian maximum ice margin is limited to that visible in first-generation geophysical data (Joon et al., 1990;

Laban, 1995; Oele, 1971) and re-processed seismic data originally acquired to map reservoir-geological features at much greater depth (Moreau et al., 2012).

A wealth of new offshore geophysical data targeting the shallow subsurface (~0 to 100 m below seafloor) has become available following significant investment into wind energy in the North Sea region. The high-resolution and high-density datasets from windfarm site investigations , provide a stimulus into new late Quaternary submerged landscape research (e.g. Cotterill et al.,

2017; Eaton et al., 2020; Emery et al., 2019; Mellett et al., 2020). In the Dutch sector of the North Sea, two new windfarm sites, Hollandse Kust Noord and Zuid, are located within previously mapped limits of the late Saalian ice sheet (Fig. 1). We integrate recently acquired high-resolution 2D seismic reflection, borehole and cone penetration test (CPT) datasets to investigate the sedimentary, geomorphological and glaciotectonic records left by the late Saalian glaciation offshore the Dutch coast. We use this data to constrain the maximum extent and morphology of the ice sheet and provide insights into the style

and dynamics of the ice sheet during the phases of ice-sheet advance, maximum extent and retreat.

Earth **Surface** Dynamics

Discussions

**Figure 1. Location of the study area in the southern North Sea showing the Hollandse Kust Noord (HKN) and Zuid (HKZ) windfarm areas. The offshore maximum ice-sheet extension for the Weichselian (Marine Isotope Stage [MIS] 2), late Saalian (MIS 6; Drenthe Substage) and Elsterian (MIS 12 and/or 10) glaciations is displayed (modified from Batchelor et al., 2019; Graham et al., 2011;**
**Hijma et al., 2012; Joon et al., 1990; Laban, 1995; and Moreau et al., 2012). The seismic tracklines displayed correspond to the multi-channel sparker data. Onshore and offshore features compiled from Laban and van der Meer (2011), Joon et al. (1990) and Peeters et al. (2016). Modern bathymetry and topography from GEBCO (https://www.gebco.net).**



## 2 Regional setting

The Hollandse Kust Noord [HKN] and Zuid [HKZ] windfarms are positioned near the southwestern rim of the Cenozoic North

Sea Basin (Van Balen et al., 2005; Ziegler, 1994). Its present-day structural configuration is the result of Late Jurassic–Early

Cretaceous rifting, followed by thermal cooling and subsidence (Glennie and Underhill, 2009; Zanella and Coward, 2003). Up

to 3500 m of Cenozoic sediment has accumulated in the central part of the basin (Knox et al., 2010). Increased Quaternary

subsidence has allowed a thick sedimentary succession to accumulate (up to 1250 m thick) (Arfai et al., 2018; Lamb et al.,

2018). The Quaternary record thins towards the southern edge of the basin reaching a thickness of 400 to 800 m at the location

of the HKN and HKZ windfarms (Cameron et al., 1984; Knox et al., 2010). The Quaternary record primarily consists of stacked

fluvial (primarily Eridanos and Rhine) and shallow marine (water depths up to 300 m, Kuhlmann, 2004) sediments with

multiple intercalations of glacial sediments.

The North Sea area experienced several major glaciations during the late Quaternary, of which the Elsterian (MIS 12 and/or

10), late Saalian (MIS 6) and late Middle Weichselian (MIS2) were the most extensive (Busschers et al., 2008; Cameron et al.,

1992; Caston, 1979; Ehlers and Gibbard, 2004; Gandy et al., 2020; Graham et al., 2011; Laban and van der Meer, 2011; Lauer

and Weiss, 2018; Lee et al., 2012; Zagwijn, 1973, 1983). In the Netherlands, the late Saalian ice sheet reached furthest south

(Fig. 1). A complex assemblage of glaciogenic sediments and glaciotectonic structures (ice-pushed ridges) surrounding tongue-

shaped basins have been identified and studied in the northern and central onshore Netherlands (Bakker, 2004; Bakker and

Van der Meer, 2003; Beets et al., 2006; Busschers et al., 2008; Van den Berg and Beets, 1987). Offshore , recorded evidence

is scarce, and the reconstruction of the late Saalian ice-sheet maximum extent is based on the study of sediments recovered in

limited boreholes and landforms imaged by seismic reflection methods (Graham et al., 2011; Joon et al., 1990; Laban, 1995;

Laban and van der Meer, 2011). Large tongue-shaped subglacial basins, smaller subglacial meltwater channel features, and in

a few exceptional circumstances, glaciotectonic deformations are present in previous generation low-resolution geophysical

data (Joon et al., 1990; Laban, 1995; Oele, 1971). West of the Dutch coastline these have been used to define the

southwesternmost Saalian ice-sheet limit near 3.5°E/52.5°N (Fig. 1, Joon et al., 1990). The presumed limits then run parallel

to the Dutch coastline (roughly N-S orientation) and eventually continue westward to the British sector along curved



Earth **Surface**
**Dynamics**
Discussions

EGU

trajectories north of 53°N (Fig. 1, Gibbard et al., 2009; Graham et al., 2011; Laban and van der Meer, 2011; Lee et al., 2012; Moreau et al., 2012).

## 3 Study area, materials and methods

### 3.1 Study area

The study area lies ~20 km west of the coastline of the Netherlands (Fig. 1) and comprises two windfarm sites, Hollandse Kust Noord (HKN) and Hollandse Kust Zuid (HKZ). The combined area of both sites is ~480 km², both are located in shallow waters of less than 30 m and the distance between both windfarm sites is 17 km. Previous research (Joon et al., 1990) suggests that HKN is positioned inside the maximum Saalian ice limit while HKZ is located on or just beyond the limit. In addition, the 105 two windfarm sites are located between two tongue-shaped glacial basins, a smaller one east of the study area (Bergen basin, Fig. 1) and a much larger one west of the study area (P/Q-block basin, Fig. 1).

### 3.2 Materials

Geophysical surveys of the HKN and HKZ windfarm development areas were carried out by Fugro between 2016 and 2018 on behalf of The Netherlands Enterprise Agency (RVO). The datasets acquired during these surveys are publicly available and 110 stored by RVO (www.offshorewind.rvo). The data include both geophysical, geotechnical and geological investigations from several phases. The shallow subsurface is imaged using sub-bottom profiler (SBP), single-channel sparker (SCS) and high-resolution multi-channel sparker (MCS). The line spacing is variable for each system and site. In HKN, the grid line spacing is 100 m in the NE-SW orientation and 2000 m in the NW-SE orientation for SBP and SCS, and 500x500 m for MCS. In HKZ, the grid line spacing is 100 m in the NE-SW orientation and 2000 m in the NW-SE orientation for SBP, and 300 m in the NE-115 SW orientation and 750 m in the NW-SE orientation for SCS and MCS. Application of tidal corrections for the three systems gives depth relative to the lowest astronomical tide (LAT). Both the SCS and MCS seismic reflection data have been depth converted by Fugro. A velocity of 1600 ms⁻¹ was used for the SCS to convert from two-way travel time (TWT) to depth in metres (m bLAT), and the MCS was depth-converted using a velocity field derived during seismic processing (Fugro, 2017a, 2016a, 2016b, 2016c, 2016d). The maximum vertical resolution is 0.7-1.8 m for MCS and 0.5 m for SCS, both with a mean 120 horizontal resolution of 2 m (Fugro, 2017a, 2016a, 2016b, 2016c, 2016d).

Earth **Surface**
**Dynamics**
Discussions

The geotechnical investigations included a combination of boreholes (BHs), downhole sampling, and/or *in situ* testing through cone penetration tests (CPTs) and targeted depths of 3 to 80 m. Boreholes were conducted using open-hole rotary drilling. Twenty-eight locations were drilled and sampled in HKN, and 35 in HKZ (Fig. 1). CPTs completed in 66 and 137 locations in HKN and HKZ (Fig. 1), respectively. Data from boreholes is limited to the borehole logs, sample descriptions and

photographs provided in reports by Fugro (Fugro, 2019, 2017b, 2017c, 2016e, 2016f). These reports also include results from pollen analyses of a selection of boreholes (StrataData, 2019, 2017). Pollen sampling was low resolution, with only a few samples per borehole, and analyses were conducted on samples that ranged from 5 to 60 cm of sediment.

### 3.3 Methods

The RVO-provided geophysical and geotechnical datasets are used in this study to investigate the landscape left by the late

Saalian glaciation. Interpretation of seismic facies and units was conducted in IHS Kingdom software according to the basic principles of seismic stratigraphy (Mitchum, 1977; Mitchum et al., 1977; Mitchum and Vail, 1977). Seismic facies were characterised following Mellett et al. (2013) and Mitchum et al. (1977) and were used to define seismic stratigraphic units. Key horizons were mapped, interpreted and gridded to maps using the flex gridding algorithm in Kingdom Suite. Infill between key horizons was characterised by their seismic facies (Table 1) and architecture, tied to relevant boreholes and CPT profiles

in key locations. The descriptions of the boreholes were cross-checked with photographs and improved where possible to construct detailed measured sections. Where available, the pollen and palynomorph data generated as part of the Fugro investigations were integrated into the analysis to cross-verify and support palaeo-environmental interpretations. The low-resolution sub-sampling for palynological assessment in the RVO-commissioned surveys prevented a detailed correlation with established chrono-biostratigraphic frameworks.





**Table 1. Summary of seismic facies identified in the study area.**

| Seis. facies | Units | Amplitude | Frequency | Continuity | Reflector termination | Structure or fill | Image |
|---|---|---|---|---|---|---|---|
| sf1 | U1 | Low | Low | Low | Not applicable | Structureless, transparent to semi-transparent | |
| sf2 | U1 | Low to medium | Low | Low | Truncation | Sub-parallel to wavy | |
| sf3 | U2 | Low | Low | Low | Onlap and truncation | Semi-transparent with sporadic reflectors | |
| sf4 | U1, U2, U3 | High | Medium | High | Not applicable | Parallel | |
| sf5 | U3 | Medium to high | Medium | High | Base: onlap Top: truncation | Parallel, channelized | |
| sf6 | U1 | Low | Low | Low | Top: truncation | Semi-transparent, dipping reflectors, trusted | |
| sf7 | U2 | Medium | Low to medium | Medium to high | Base: onlap, downlap Top: truncation | Channel complex (multihistory) | |
| sf8 | U2 | Medium | Low | High | Base: downlap Top: truncation | Oblique-tangential clinoforms | |
| sf9 | U1, U3 | Low to medium | High | High | Truncation | Parallel, drape | |
| sf10 | U3 | Low | Variable | Low | Onlap | Sub-parallel, sheet | |

Earth **Surface**
**Dynamics**
Discussions

## 4 Results and interpretation

Three seismic stratigraphic units (U1, U2 and U3) were defined by mapping two key surfaces (S1 and S2) based on the
integration of geophysical and geotechnical datasets from both windfarm sites. The seismic unit distribution varies spatially,
from north to south across the study area (Fig. 2). The correlation of seismic units between both windfarm sites is based on the
nature of their bounding surfaces, their sedimentological characteristics and pollen content. We have also examined available
legacy datasets between both windfarm sites, but they showed limited resolution or penetration.

The lower seismic unit (U1) is present throughout the study area. We do not distinguish a basal contact to U1 (Fig. 2). In
general, U1 is characterised by a weak acoustic response, and its internal structure is distinguished only locally in its upper
part, comprising sub-parallel, low frequency, low-to-medium amplitude reflectors that are truncated by disconformity S1.
Seismic unit 2 (U2) is only present in the north of the study area, infilling depressions of S1 (Fig. 2). U2 seismic facies are
generally transparent, but some low amplitude reflectors are distinguished to the south onlapping and downlapping the
underlying U1. The top of U2 is defined by disconformity S2, which truncates underlying units across the study area (Fig. 2).
The overlying seismic unit (U3) thickens southward, where it lies directly over U1, and S1 and S2 are coincident. The seismic
facies of U3 are highly variable. Both U1 and U3 comprise several stratigraphic units, but remain undifferentiated in this study
for simplicity.

Marked spatial differences in both the internal structure of the seismic units and the morphology of their bounding surfaces
exist across the study area (Fig. 2). Hence, we present detailed descriptions geographically and consider three sectors: northern
sector of HKN, middle sector of HKN, and southern sector of HKN and the HKZ windfarm, integrating both the geophysical
(seismic) and geotechnical (sampled boreholes, CPTs) datasets (Fig. 3).

**Figure 2. Representative seismic profiles (multi-channel sparker) and interpretation panels across Hollandse Kust Noord (HKN,**
**above) and Hollandse Kust Zuid (HKZ, below) windfarm areas showing the main seismic stratigraphic units and surfaces. The**
**separation distance between the northern (HKN) and southern (HKZ) profiles is 17 km. For location of the seismic profiles see the**
**inset map.**







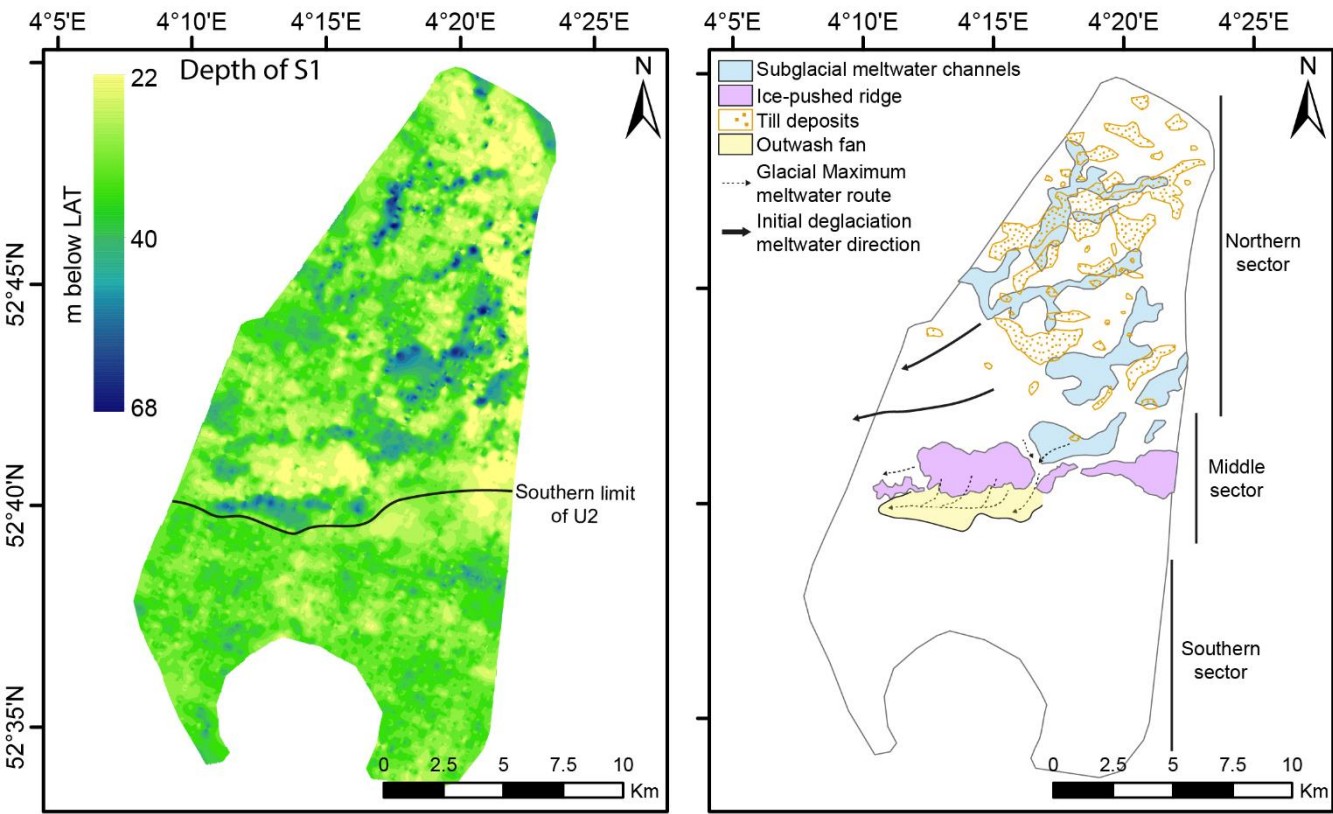

**Figure 3. Seismic mapping and geological interpretation of HKN windfarm site. Left: Map of the depth of surface S1 (base of unit U2, see also Figs. 2 and 4). Right: Landform interpretation showing subglacial meltwater channels, ice-pushed ridges, till deposits and an outwash fan mapped in the study area. Meltwater direction during the early deglaciation has been inferred from the general relief and slope of surface S1 and the distribution of glaciofluvial deposits.**

## 4.1 Northern sector of HKN

### 4.1.1 Description

In the northern sector of HKN, the basal seismic U1 is generally characterised by a weak acoustic response, displaying transparent and semi-transparent facies (sf1, Table 1). Locally, discontinuous slightly-wavy or sub-parallel reflectors are observed in the upper part of the unit, truncated by S1 (sf2, Table 2). Most boreholes are located where U1 predominantly displays transparent acoustic facies and comprise silty fine-to-medium sand. Rare dark organic clay and peat are recovered at deeper levels (Fig. 4).



**Figure 4. Stratigraphic section (multi-channel sparker) through subglacial meltwater channels in the northern sector of HKN windfarm area. Borehole HKN10 was recovered in one of these channels capturing the sedimentary characteristics of the three main seismic stratigraphic units. Only the upper 50 m of the borehole are shown. Cone resistance from a CPT obtained in the same site is displayed on the side of the borehole and in the seismic profiles. For location of the seismic profiles and the borehole see the inset map (same map from Fig. 3). Grain size: C – clay, St – silt, fS – fine sand, mS – medium sand, cS – coarse sand, G – gravel. Vertical dashed line indicates the position where the lines cross each other.**

Earth **Surface**
**Dynamics**
Discussions

In the northern sector, S1 is characterised by deep V-shaped incisions filled by U2 (Fig. 4). The detailed mapping of S1 in the HKN windfarm site shows a set of elongated depressions, generally oriented NE to SW, and up to 68 m deep (m below LAT,

Fig. 3). These depressions are highly irregular in shape, of varied dimensions (0.5 to 8 km long and up to 1 km wide) and are characterised by steep walls (5-20°), an undulating thalweg and abrupt terminations. U2 infills the depressions and is characterised by semi-transparent facies and discontinuous sub-parallel reflectors (sf3, Table 1, line A-A', Fig. 3). A set of three high-amplitude discontinuous reflectors are commonly observed at the base of U2 (sf4, Table 1), corresponding in the boreholes to an admixture of clay, silt, sand and gravel (clay-rich diamicton, Fig. 4), and in CPT logs by low cone resistance

values due to the high clay content. The overlying infill corresponds to very dense medium-grained sands, sometimes containing thin laminae of silt and clay (Fugro, 2019). Pollen analyses performed in borehole HKN10 (StrataData, 2019) indicate relatively high numbers of fern and moss spores and freshwater algae, while tree pollen is low and dominated by *Pinus*. Towards the top of U2 (10-12 m in HKN10), moderate numbers of other tree types are also reported, mainly *Alnus*, *Betula*, *Corylus* and thermophilous *Quercus* (StrataData, 2019).

Seismic unit U3 corresponds to the depositional package that onlaps U2 and U1. In general, U3 is thin (<5 m) displays a sheet-like geometry, and locally thickens above concave-up isolated depressions (e.g. section A-A' in Fig. 4). Seismic facies of U3 are more distinct on SBP data, displaying parallel, high to medium amplitude reflectors (sf5, Table 1). In boreholes, this unit mainly corresponds to heterolithic interbedded sands and silty sands with abundant shells and shell fragments. These deposits are characterised by relatively high numbers of foraminiferal test linings and dinocysts (StrataData, 2019). Where basal

depressions are present, sands overlay dark organic, laminated clayey silts (Fig. 4). Laminated silts recovered at the base of U3 in borehole HKN10 (Fig. 4) are dominated by tree pollen, mainly *Quercus*, *Pinus*, *Alnus*, *Corylus* and *Betula* (StrataData, 2019).

### 4.1.2 Interpretation

The most prominent features in the northern sector of the study area are the deep NE-to-SW trending depressions that truncate

U1 (Figs. 3 and 4). Their abrupt terminations, the rising and falling thalweg, and the lack of connection between the different depressions means they are unlikely to represent a fluvial channel network. Instead, they are consistent with the features of subglacial meltwater channels and tunnel valleys (Kehew et al., 2012; van der Vegt et al., 2012), which typically form during

Earth **Surface**
Dynamics
Discussions

phases of ice advance. The disrupted aspect of reflectors from the underlying seismic unit (U1, sf2) is interpreted as

glaciotectonic deformation of the pre-existing deposits due to ice loading (Phillips et al., 2018). The patchy high-amplitude

reflectors found at the base of the subglacial meltwater channel-fills, corresponding to matrix-supported diamicton in the

boreholes, are interpreted as glacial tills. These deposits are restricted to the northern sector of HKN and are associated with

the subglacial meltwater channels and their margins.

The upper part of U2 is characterised by a weak acoustic signal and occasionally sub-parallel reflectors, mainly corresponding

to dense fine and medium sands. The freshwater algae content and the dominance of *Pinus* pollen indicate deposition under

cool or cold conditions in a terrestrial or freshwater setting, although the presence of thermophilous species, such as *Quercus,*

towards the top U2, suggests a progressive transition towards a warmer climate. Given the seismic architecture, these sediments

are likely pro- or subglacial meltwater channel-fills that formed during ice retreat.

Boreholes through U3 are characterised by an increase in marine indicators (shells, dynocysts, foraminifera) and tree

pollen is abundant at the base (e.g. *Quercus*, StrataData, 2019), indicating mild climatic conditions. Overall, U3 aggrades and

drapes older deposits and is interpreted as coastal and/or marine sediments deposited during marine transgression.

### 4.2 Middle sector of HKN

### 4.2.1 Description

The middle sector of HKN windfarm shows local intense deformation where the reflectors of U1 are distorted (Fig. 5). In this

area, the top of U1 occurs < 5 m beneath the seafloor. Although the acoustic signal of U1 is weak, some steeply dipping

reflectors (4-15°) are distinguished (sf6, Table 1) that show evidence of intense, southwards verging thrusts (Fig. 5). Mapping

this zone of contractional deformation reveals an E-W oriented ridge-like feature denoted by linear topographic highs in the

contour map of S1 (Fig. 3). Seismic profiles through this area (Fig. 2 and 5) revealed that the thrusted ridge is up to 4 km wide,

buried beneath the younger U2 and U3 deposits. Borehole HKN56 is sited on top of this ridge (Fig. 5). Sediments from U1

correspond to laminated sandy clays (from 34 to 39.5 m) with interbedded sand and peat overlain by a thick, massive deposit

of dense, silty-fine sand (from 4 to 34 m, HKN56, Fig. 5). Clay and peat beds appear laterally extensive as they are intersected

by several boreholes between 36 and 42 m below the seafloor in the middle of HKN. These beds are highlighted in CPT



profiles by low cone resistance values and a high friction ratio. These clay and peat deposits are found at the base of the

thrusted blocks (Fig. 5), delimiting a ~35 m thick package of deformed strata (Fig. 5).

**Figure 5. Stratigraphic sections (multi-channel sparker) through the glacio-tectonic ridge in the middle sector of HKN windfarm area. In seismic section C-C', the top of the ridge is incised by proglacial meltwater channel fills. Borehole HKN56 was recovered at**

**the top of the ridge. Only the upper 50 m of the borehole are shown. Cone resistance from a CPT obtained in the same site is displayed on the side of the borehole and in the seismic profiles. In section D-D', outwash sediments (outwash fan) prograde inside a shallow depression incised in the south flank of the ridge. For location of the seismic profiles and the borehole see the inset map (same map**
**from Fig. 3). Grain size: C – clay, St – silt, fS – fine sand, mS – medium sand, cS – coarse sand, G – gravel.**

The thrusted ridge segment is truncated at the northern flank by an E-W trending incision at the base of U2 (Figs. 3 and 5).

Here, seismic facies of U2 are of variable amplitude and frequency, generally with onlapping subparallel reflectors (sf7, Table

1). Several phases of incision and infill can be identified (section C-C', Fig. 5). To the south of the ridge, there is a large E-W-

trending depression (Figs. 3 and 5). However, the base of this depression is mainly sub-horizontal and its relationship with

underlying strata is not clear. This depression is filled by a set of clinoforms with a southwards progradational trend (Fig. 3)

that are characterised by medium to high amplitude, oblique-tangential reflectors (sf8, Table 1, Fig. 5). U1 and U2 are overlain

by U3, which in borehole HKN56 corresponds to laminated fine to coarse-grained sands containing abundant bivalve shells

and shell fragments (Fig. 5).

**4.2.2 Interpretation**

The prominent ridge of thrusted and folded strata (Fig. 3), with highly deformed sediments of U1 preserved a few meters

beneath the modern seafloor, is consistent with glaciotectonised sediments marking an ice-sheet advance into this part of HKN.

The glaciotectonised segment is up to 4.5 km wide and glaciotectonic disturbance affects deposits down to 30-40 m below the

seafloor, where the extensive peat and clay layers likely form a basal detachment surface (décollement).

The primary E-W direction of the depression that truncates the north flank of the glaciotectonised unit contrasts with the

dominant NE-SW direction of the subglacial channels identified in the northern sector of the windfarm (Fig. 3). The acoustic

character of U2 here is also different, displaying sub-parallel reflectors and a multiphase history of infill (line C-C', Fig. 5).

Our preferred interpretation is that these incisions are proglacial meltwater channels formed during a phase of ice retreat (post-

glaciotectonic) that were probably filled by glacial outwash sediments as meltwater discharge eroded and reworked the top of

the glaciotectonised ridge and surrounding areas. The interpretation of the depression on the southern side of the glaciotectonic

ridge is less clear. It could correspond to an incision due to the erosive action of meltwater or to a syncline that was overthrusted

from the north and later infilled with outwash deposits. The location, morphology, and internal structure of the southward

progradational clinoforms accumulated within this depression supports an interpretation as an outwash fan or a fan-delta that

prograded into a proglacial lake. Glaciotectonised and glaciofluvial deposits in this area are buried by a drape of coastal and marine sands (U3, Fig. 5), which thickens to the south (Fig. 2), similar to U3 in the northern sector of HKN.

### 4.3 Southern sector of HKN and the full HKZ windfarm

**4.3.1 Description**

We group the southern sector of HKN windfarm and the whole of the HKZ windfarm area together because of their similarity in seismo-stratigraphic architecture, which is characterised by the southward thickening of U3 that is directly overlying U1. U2 is absent and S1 and S2 merge into a single unconformity (Fig. 2). U1 in this sector is generally characterised by more acoustically stratified facies, although the acoustic return is still weak. Parallel horizontal reflectors of low amplitude and high frequency (sf9, Table 1) are common in the upper part of the unit, while transparent or semi-transparent facies are dominant at depth (sf1, Fig. 6). Locally in the HKZ windfarm, some discontinuous reflectors of very-high amplitude (sf4) are identified in U1 (Fig. 7). In the boreholes, the lower part of U1 corresponds to laminated fine- to medium-grained sand containing some interbedded peat (boreholes HKN72 and HKN70, Fig. 6, and borehole HKZ4-BH04, Fig. 7), while the upper part of U1 comprises laminated fine- to medium-grained sands with some thin laminae of clay, and interbedded sandy clays and silty sands. This is reflected in the CPT response, displaying a highly variable (serrated) cone resistance for interbedded facies. Pollen analyses of samples from the lower part of U1 in boreholes HKN70, HKN72 and HKZ4-BH04 (StrataData, 2019, 2017) show a consistent presence of tree pollen, including *Pinus*, *Alnus*, *Quercus*, *Ulmus* and *Tilia*, with low to moderate numbers of non-tree pollen, spores and freshwater algae. Marine content is minimal and may be reworked. In the upper part of U1, *Pinus* dominates pollen assemblages, with low to moderate numbers of other tree pollen, non-tree pollen, herbaceous pollen and fern spores. Marine taxa are abundant, including dynocysts (such as *Spiniferites* spp.) and foraminiferal test linings (StrataData, 2019, 2017).

Earth **Surface**
Dynamics
Discussions

**Figure 6. Stratigraphic sections (multi-channel sparker) in the southern sector of HKN windfarm area where unit U3 is directly lying on unit U1 (i.e. surfaces S1 and S2 seismically inseparable). Only the upper 50 m of the boreholes are shown. Cone resistance from CPTs obtained in the same sites is displayed on the side of the boreholes and in the seismic profiles. For location of the seismic profiles and the borehole see the inset map (same map from Fig. 3). The southern limit of unit U2 is also displayed on the map. Grain size: C – clay, St – silt, fS – fine sand, mS – medium sand, cS – coarse sand, G – gravel.**



U3 in this sector is characterised by sub-horizontal discontinuous reflectors, generally of low amplitude and variable frequency,

and a sheet-like geometry. A set of 3 to 4 very high amplitude reflectors (sf4) are sometimes observed at the base of U3 (e.g. section F-F' of Fig. 6, and Fig. 7) and occasionally at shallower depths (e.g. section B-B', Fig. 7). Several slightly erosive surfaces truncate reflectors, and channel-like incisions are observed within the unit (Fig. 7). Boreholes in HKN record up to 10 m of laminated fine- to medium-grained sand and silty sand containing some shell fragments (Fig. 6). In HKZ, U3 is thicker (~25 m in borehole HKN4-BH04) and comprises laminated clays with some thin laminae of sand at the base of the unit, passing

upward to medium-grained sands containing abundant shell fragments (Fig. 7). The very high-amplitude reflectors recorded in the seismic data (sf4) correlate to peat layers of variable (~0.2-2 m) thickness (e.g. in boreholes HKN70 and HKZ4-BH04, Figs. 6 and 7). Pollen recovered in the clay and peat at the base of U3 (10-12 m in HKN70 and 20-25 m in HKZ4-BH04) are characterised by an increased presence of tree pollen, with high numbers of *Pinus*, *Quercus*, *Alnus* and *Ulmus* (StrataData, 2019, 2017). Fern and moss spores are also frequent and marine taxa, such as dynocysts, are minimal but present. In borehole

HKN70, these deposits are overlain by sands dominated by *Pinus* (StrataData, 2019), which pass upwards to modern shelly marine sands (Fig. 6). Cold and warm tree pollen assemblages alternate up borehole HKZ4-BH04 (10-20 m, StrataData, 2017) with some intervals of increased reworking and marine influence (Fig. 7).

### 4.3.2 Interpretation

In this area, both U1 and U3 are characterised by a sheet-like geometry and bounded by a smooth and slightly erosive surface

(S1+S2) without prominent incisions. Deposits of U1 generally display parallel to sub-parallel reflectors. Deposits from the lower part of U1 contain interbedded peat, abundant tree pollen (including *Quercus*) and minimal marine content, pointing to deposition in a terrestrial setting under mild climatic conditions. In the upper part, the interbedded sands and clays and the abundant marine taxa support the interpretation of these deposits as a near-coastal setting, probably deposited under cooler climatic conditions as the pollen assemblages are dominated by *Pinus*.







**Figure 7. Representative seismic profiles showing the stratigraphy underlying the HKZ windfarm site. Section A-A' corresponds to a multi-channel sparker line, section B-B' to a single-channel sparker line. Borehole HKZ4-BH04 illustrates the sedimentological characteristics of units U1 and U3 in this windfarm. Only the upper 50 m of the borehole are shown. Cone resistance from a CPT obtained in the same site is displayed on the side of the borehole and in the seismic profiles. For location of the seismic profiles and the borehole see the inset map. Grain size: C – clay, St – silt, fS – fine sand, mS – medium sand, cS – coarse sand, G – gravel. Vertical dashed line indicates the position where the lines cross each other.**

Earth **Surface**
**Dynamics**
Discussions
EGU
The internal structure of U3, characterised by sub-horizontal reflectors displaying a generally aggrading pattern, indicates that these sediments were deposited during a phase of increasing accommodation. However, the existence of several internal erosion surfaces (Fig. 7) suggests that these deposits are not the result of a single phase and point to a complex depositional

history. The cyclicity of the pollen successions and the complicated internal structure, with several intercalated levels of peat and multiple lower-order disconformities (Fig. 7), suggest that this unit probably records multiple glacial-interglacial cycles of which mainly coastal and shallow marine deposits were preserved.

## 4.4 Seismo-stratigraphic and lithostratigraphic interpretation

Integration of geophysical and geotechnical datasets offshore the Dutch coast allows identification of a major unconformable

surface (S1) formed by ice-driven erosion. The analysis also points a single glacial episode to have created this glaciogenic unconformity. Connecting to lithostratigraphic and chronostratigraphic frameworks established for the adjacent onshore (Fig. 1), we equate the disconformity S1 to the Drenthe glaciation episode of the Saalian glacial stage (ca. 160 ka; mid MIS 6), in line with earlier ice-limit studies covering the southern North Sea (Fig. 1, Gibbard et al., 2009; Graham et al., 2011; Joon et al., 1990; Laban and van der Meer, 2011; Moreau et al., 2012).

U1, which is irregularly truncated by S1, spans deposition up to the Drenthe glaciation. The seismic facies of U1, which is mostly transparent and semi-transparent in nature, but displays low-amplitude sub-parallel reflectors towards the top, suggests several multiple lithostratigraphic formations. The upper part of U1 is interpreted to correspond to the late Middle Pleistocene (MIS 7 or older) marine Egmond Ground Fm. (Cameron et al., 1992, 1984; Oele, 1971; Rijsdijk et al., 2005; TNO-GSN, 2021) given the presence of marine taxa (Figs. 6 and 7). The lower part of U1 likely corresponds to the early Middle Pleistocene

Yarmouth Roads Fm., which is regionally extensive throughout the southern North Sea (Cameron et al., 1984). The Yarmouth Roads Fm. (corresponding to Formation 4.1.1 in Rijsdijk et al., 2005), includes both marine and fluvial facies that mainly consists of fine or medium-grained non-calcareous sands, with variable clay lamination and local intercalations of reworked peat (Cameron et al., 1984; Rijsdijk et al., 2005; TNO-GSN, 2021). The weak acoustic character of U1 hinders the clear identification of a bounding surface between both formations.

U2 is confined to subglacial meltwater channels and proglacial and deglacial outwash complexes, directly overlying the late Saalian glaciogenic unconformity (S1, Fig. 4). Such glacial and glaciofluvial sediments are assigned to the Drenthe Fm., which



includes all glaciogenic deposits of this glaciation episode (Rijsdijk et al., 2005; TNO-GSN, 2021). Till deposits at the base of U2 correspond to the Gieten Member and the overlying glaciofluvial deposits to the Schaarsbergen Member in this formation (Rijsdijk et al., 2005; TNO-GSN, 2021). Chronostratigraphically, these sediments accumulated in the middle to late Saalian

(second half of MIS 6); attributed a numeric age from ca. 155 ka onwards following OSL dating in Busschers et al. (2008). This would be time-equivalent with Drenthe-II and Warthe Substages recognised some 250 km to the northeast of our study area (Elbe estuary and German Bight; e.g. Ehlers and Gibbard, 2004).

U3 comprises a complex internal architecture with several minor-order disconformities. Laminated clays and sands, peat beds, and modern shelly marine sands are included in this unit. Basal pollen assemblages point to a warm interglacial or interstadial,

dominated by tree pollen with high numbers of *Quercus* (oak), known to have first re-established in the southern North Sea in the Eemian, which set on considerable time after the Drenthe Substage deglaciation (after 130 ka, within MIS 5e). Fluctuating pollen assemblages suggest that U3 records more than one climatic cycle. We interpret that deposition of U3 started during Termination II (MIS 6/5 transition), from the onset of the Eemian interglacial onwards. U3 appears to have recorded climatic oscillations through the Eemian, the Weichselian and into the Holocene, explaining its complicated internal structure. It is

topped by the modern seafloor. Future detailed analysis of this unit, using higher-resolution seismic profiles, may allow the differentiation of lithostratigraphic formations.

## 5. Discussion

### 5.1 Glaciotectonic structures and the maximum late Saalian ice-sheet extent

The sedimentological, seismic stratigraphic and geomorphic analyses performed in the study area revealed the preservation of

a glacial landscape offshore the Dutch coast. It is characterised by the presence of subglacial meltwater channels and a glaciotectonic ridge in HKN, that can be classified as a composite ridge or thrust-block (push) moraine (<100 m or relief, Aber et al., 1989; Aber and Ber, 2007; Van der Wateren, 2002, 1995, 1985), which are usually associated with proglacial or sub-marginal glaciotectonics, forming parallel to the ice margins, marking glacier stillstands or readvances (Aber and Ber, 2007; Benn and Evans, 2010).

Earth **Surface**
**Dynamics**
Discussions

Some of the most studied late Saalian composite ridges are found in the onshore Netherlands, at the edges of tongue-shaped

glacial basins (Fig. 1), and by comparison are generally large, with a total relief exceeding 200 m. The Veluwe ridge in the

west side of the IJssel glacial basin (Fig. 1) comprises a ~200 m thick package of unconsolidated deformed strata, displaying

staked thrusts close to the ice-contact that transitions into folded strata in the distal areas (Bakker, 2004; Bakker and Van der

Meer, 2003; Van der Wateren, 1995). The structural style of deformation in the Veluwe ridge is different to that found in HKN.

The features mapped in HKN are similar in nature to the thinner (60-80 m) onshore Utrecht Ridge in the southwestern margin

of the Amersfoort basin (Fig. 1) (Aber et al., 1989; Aber and Ber, 2007). This ridge is composed of imbricated thrust blocks

of unconsolidated Pleistocene strata striking parallel to the ridge in a package of deformed sediments. The Utrecht ridge is

about 2.5 km wide, displays a plateau top, and is slightly asymmetric in cross profile as the southwestern flank grades into a

sandur outwash complex. According to van der Wateren (1995), the style of glaciotectonic deformation of the Utrecht ridge is

dominated by thrusting. Although the top plateau has been flattened, this is consequence of denudation and erosion, and not

due to the ridge being overridden by the ice sheet. If it had been overridden, an overprint of the structural style by subglacial

deformation (compressive and extensional) would be expected (Van der Wateren, 1985). Considering the deformation style

and geomorphology, the ridge in HKN likely represents an ice-marginal position denoting the ice-sheet maximum extent,

similar to the Utrecht ridge.

Similar features are seen in the wider southern North Sea. The dimensions, seismic structure and depth of the ridge in HKN

are also comparable to the Weichselian thrust-block moraines described in the Dogger Bank area (Cotterill et al., 2017; Emery

et al., 2019; Phillips et al., 2018) and the Elsterian composite ridges identified in the Dudgeon windfarm offshore north Norfolk,

UK (Mellett et al., 2020). Both cases are examples of well-preserved glaciotectonised packages up to 40-50 m thick (bounded

by a basal detachment surface visible in seismic profiles) composed by thrusted and folded unconsolidated sediments. In the

Dudgeon windfarm and the eastern sector of Dogger Bank, the glaciotectonised sequences form multiple parallel ridges

extending over several kilometres, which are interpreted to be the result of active ice-sheet retreat (Mellett et al., 2020; Phillips

et al., 2018). In the western sector of Dogger Bank, there is a single ridge bounding an area of subglacial meltwater channels

and streamlined bedforms (lying to the north of the ridge), interpreted as a terminal thrust-block moraine denoting the

maximum extent of the British-Irish ice sheet during the Weichselian (Emery et al., 2019).

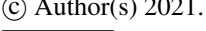

It is reasonable to conclude that the shallow occurrence of U1 in the HKN windfarm occurred in response to thrusting in front

of an advancing ice sheet. Given the absence of a clear palaeo ice-sheet signature in sediments preserved further to the south,

this glaciotectonised land system likely records the maximum southernmost position of the late Saalian ice sheet (Fig. 8). This

contrasts with the location of the MIS 6 ice-sheet limit which has previously been mapped ~50 km south, through the central

part of the HKZ (Fig. 8, Joon et al., 1990; Laban, 1995; Laban and van der Meer, 2011), based on indirect evidence observed

in legacy seismic profiles.  These mainly comprised of south and west-dipping reflectors imaged a few meters beneath present-

day seafloor (Joon et al., 1990; Laban, 1995), which were inferred to represent ice-pushed ridges, in continuation to those

present on land (Fig. 1). This interpretation is not supported by the newly acquired geophysical and geotechnical datasets,

which provide higher resolution imaging of the shallow subsurface and show no unequivocal evidence of glaciotectonic

deformation in the HKZ windfarm area (Figs. 2 and 7). Fugro (2017c) also reported a structure in the HKZ windfarm site that

could potentially represent glaciotectonic deformation, a thin package of wavy north dipping reflectors. Upon careful

examination of the data, this feature is only visible in a single SCS profile, lacks a minimum lateral continuity (as it is not

present in any of the parallel profiles to the west or the east) and is associated to channel incisions and infills. Therefore, we

cannot confidently attribute this structure to glaciotectonic deformation, as it is not different from any other channel and bars

fills which are locally extensive as part of U1 in HKZ. These channel and bar deposits are probably the same structures

suggested by Joon et al. (1990) and Laban (1995) to represent ice-pushed ridges.

Our new mapping (Fig. 8) shows the ice margin in the offshore sector to be more irregular than previously presumed, displaying

a lobate shape similar to the marginal features mapped onshore (Fig. 1). The lobate shape onshore is explained by the influence

of subsurface structural and hydrological conditions, which developed due to contrasting hydraulic conductivities close to the

surface (Van den Berg and Beets, 1987; Van der Wateren, 1995, 1985). The onshore ice-pushed ridges formed in areas covered

by coarse-grained fluvial material, where a fine-grained layer in the lower substratum could act as a basal detachment, while

ice flow was channelised through the glacial basins. The distribution of the glacial basins is controlled primarily by deeper

Cenozoic tectonic structures (De Gans et al., 1987; Van der Wateren, 2003). By comparison, no drastic changes in the

subglacial bed conditions are expected offshore, as the underlying Cenozoic stratigraphy in the area is dominated by the sandy

deposits of the Yarmouth Roads Fm. (Cameron et al., 1984; Laban, 1995). The offshore lobate margin shown in Figure 8B is

Earth **Surface**
**Dynamics**
Discussions
EGU
in part determined by the proposed Saalian margin at the southwestern edge of the comparatively large P/Q-block glacial basin

(west of our study area), which was mapped using legacy seismic data (Joon et al., 1990; Laban, 1995).

**Figure 8. (A) Conceptual model of the infill (following transgression from MIS 6 to MIS 5e) showing the main glacial geomorphological features and sediments preserved in Hollandse Kust Noord. (B) Comparison between the previously inferred**
**maximum Saalian ice sheet extent (dashed line) and the new limit proposed in this study (solid line). Legend and sources as in Fig. 1. (C) Bathymetry profile through the Hollandse Kust Noord area highlighting the glacial geomorphological features identified and the approximate position of the ice front during the maximum ice sheet extent.**

We highlight an interesting difference between the ice-push ridges surrounding the glacial basins offshore versus onshore. The

ice-push ridges in HKN windfarm are oriented perpendicular to the P/Q-block basin, whereas onshore in the central

Netherlands, the ridges formed parallel to the basin rims. This suggests a glaciogenic difference in formation processes,

particularly of the P/Q-block basin offshore vs. the equally sized Amersfoort and IJssel basins to the east onshore.  The current



windfarm data coverage does not allow us to re-evaluate the Saalian margin position through the P/Q-block basin, but given

that new-high resolution geophysical data had led to re-evaluation of the margin position through HKN and HKZ, a similar

analysis may be considered to assess the presence of a lobate margin, and thereby enhance understanding of ice sheet processes.

**5.2 The late Saalian glacial and deglacial phases in the southern North Sea**

During the Drenthe Substage, Scandinavian ice advanced from Swedish and Baltic sea source areas into northern Germany

and the Netherlands from the ~NE (Ehlers and Gibbard, 2004; Van den Berg and Beets, 1987; Zagwijn, 1973). In the

Netherlands, the ice advance during MIS 6 forced the Rhine-Meuse fluvial system to a southerly (proglacial) position where

it connected with glacio-fluvial (sandur) systems in front of the ice sheet (Busschers et al., 2008). The ice sheet eventually

advanced into the area offshore the Holland coast where it triggered significant erosion, recorded in the HKN windfarm by

surface S1 (Figs. 8 and 9A). Over-pressurised subglacial meltwater carved deep V-shaped NE-SW orientated channels beneath

the ice sheet, which are well-preserved in the northern sector of HKN (Figs. 3 and 4), and indicate regional ice advance from

the northeast (Huuse and Lykke-Andersen, 2000; Moreau et al., 2012; van der Vegt et al., 2012). Large-scale glaciotectonic

deformation occurred as the ice sheet pushed into, and overrode, pre-existing sediments leading to the stacking of detached,

thrust-bound sediments (U1) in the middle sector of HKN (Figs. 5 and 8). The southern flank of the glaciotectonic ridge is

covered by a proglacial fan of outwash deposits that prograded into the nearby water-filled depression (Figs. 5 and 9A).



Earth **Surface**
**Dynamics**
Discussions

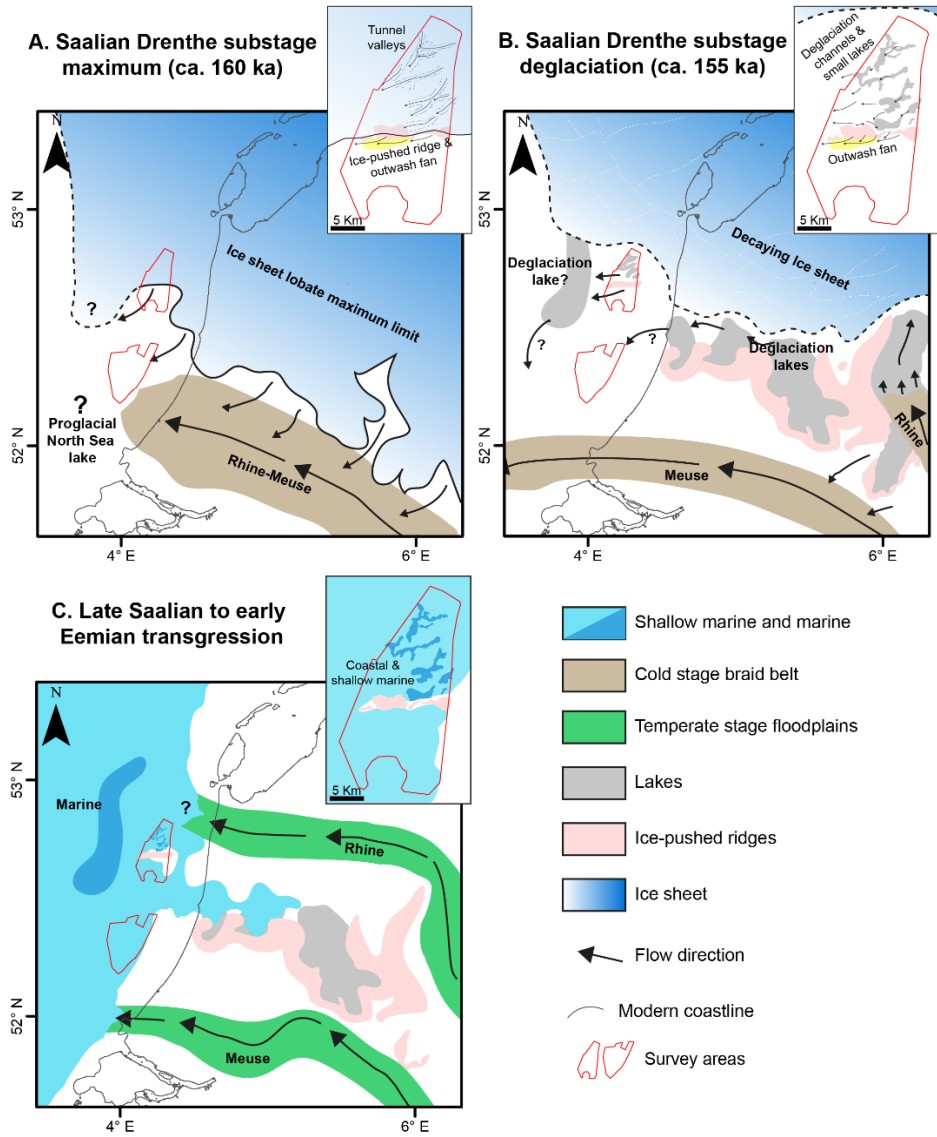

**Figure 9. Palaeogeographic maps depicting the landscape evolution in the study area. Inset maps show the detailed evolution reconstructed in Hollandse Kust Noord (HKN) windfarm site. (A) Palaeogeography during maximum ice-sheet extent in the Saalian**
**Drenthe substage ca. 160 ka (Busschers et al., 2008; Toucanne et al., 2009). Ice advancing into HKN deformed the underlying strata and carved subglacial meltwater channels. An E-W oriented glaciotectonic ridge indicates the position of the maximum ice front (this study). A contemporary proglacial lake has been proposed in earlier work (Busschers et al., 2008; Hijma et al., 2012), but no evidence/confirmation has been found in this study. (B) Saalian Drenthe substage deglaciation ca. 155 ka (Busschers et al., 2008; Toucanne et al., 2009) characterised by regional ice stagnation (dead ice). Overspilling, interconnected lakes of varied dimensions**
**occupied the deglaciated tongue basins (Beets et al., 2006; Busschers et al., 2008; Laban, 1995; van Leeuwen et al., 2000; Zagwijn, 1996, 1983), as well as smaller-scale subglacial channels (this study). The Rhine river diverted into a larger basin in the east of the depicted area. Drainage in the HKN study area was mainly directed to the larger 'P/Q' tongue basin lake in the west. (C) Latest Saalian to early Eemian North Sea transgression phase, depicted for a sea-level elevation of 30 m below present-day level. Coastal to marine transgressive deposits buried the Saalian glacial landscape in the study area (Peeters et al. 2016, 2015; Van Leeuwen et al.**
**2000; Zagwijn, 1983). The distribution of these deposits and the timing of the flooding is conditioned by the antecedent topography inherited from the late Saalian glaciation.**



Three phases of Saalian Drenthe ice advance have been postulated in the Netherlands. An earliest ice advance from the NE invaded the northern Netherlands and reached a stationary ice margin (Rappol et al., 1989), releasing englacial debris that formed marginal and fluted moraines onshore (Passchier et al., 2010; Van den Berg and Beets, 1987). Advance then

recommenced, and the maximum extent limit across the central Netherlands was reached (i.e. the identified offshore HKN ridge, the tongues carving the Haarlem and Amsterdam basins, the further tongues pushing up the Utrecht and Veluwe ridges, Figs. 1 and 9A). Besides the geographical position, also the dominant NE-SW direction of the subglacial meltwater channels preserved in HKN suggests that the ice sheet reached the study area in this main phase of ice advance during MIS 6. To the east of the study area, a final phase of readvance has followed the main advance, based on large scale NNW-to-SSE fluted till

morphology observed the northeast of the Netherlands ("Hondsrug" ice flow system; Van den Berg and Beets, 1987) and this appears to have exacerbated ice-marginal meltwater routing along the main advance ice margin (Meinsen et al., 2011). An explanation for this reorganisation in ice flow direction is a proposed change in coalescence of Scandinavian and British ice in the central North Sea (Ehlers, 1990; Kluiving et al., 1991; Rappol et al., 1989). In the HKN windfarm area, no direct evidence for change in ice-flow direction appears registered. The glaciotectonic landform preserved in HKN points to a land-based ice

sheet that overrode an exposed North Sea basin floor in a terrestrial setting where outwash deposition dominated. We find no indication in HKN or HKZ of clear water lain deposition, that could mark deposition in a hypothesised proglacial lake at the time of maximum extent (Busschers et al., 2008; Hijma et al., 2012).

Following ice retreat from HKN, subglacial channels were exposed, trapping fine-grained outwash sediments in bodies of standing water (U2; fig. 9B). Overspill channels likely connected the different pools and lakes, explaining the channel-like

incisions that are sometimes observed as concave-up reflectors in the upper part of U2 (Fig. 4). In the middle sector of HKN (Fig. 3), outwash channels erode the northern flank of the ice-pushed ridge and, locally, its upper boundary (Fig. 5). The presence of outwash deposits seems to be restricted to the north and middle sectors of HKN (Fig. 2). Only a diffuse drainage network can be inferred from mapping surface S1 (Fig. 3), and a few channelised deposits are identified, mainly oriented E-to-W, near the ice-pushed ridge. These outwash channels may have drained west into the large P/Q-block glacial basin (Joon

et al., 1990; Laban, 1995; Oele, 1971), which with its base at 66 m below sea level likely acted as an effective sink for meltwater and outwash sediments (Fig. 9B). By comparison, in the southern part of HKN and all HKZ, disconformities S1 and S2 are



coincident. It is possible that any outwash deposits here were removed/reworked in the late Saalian/early Eemian; Saalian glaciofluvial sediments that incorporated into the Eem Fm. have been reported in the southern North Sea (Laban, 1995). Any glaciofluvial outwash sediments preserved to the south may also be very thin, patchy, and difficult to identify in seismic data.

Ice retreat left an unusual landscape preserved in HKN, characterised by the progressive filling of ice-advance and glacio-fluvial drainage depressions (Fig. 3). Passchier et al. (2010) proposed a model of rapid ice retreat because of ice streaming in developing phases of deglaciation (i.e. from ca. 155 ka onwards in the southern North Sea) based upon the presence of meltwater channels incised into the floors of glacial troughs in the southern North Sea. The Hondsrug phase evidence mentioned above could alternatively be explained in this way too. Geomorphological evidence for ice streaming (Stokes and

Clark, 1999; Winsborrow et al., 2010) is absent in the HKN area and there is only evidence of main phase ice margin stagnation. Subglacial topography is also one of the strongest controlling factors that determine the location of ice streams, as topographic focusing favour fast ice flow (Stokes and Clark, 1999; Winsborrow et al., 2010). The general topography in the study area is relatively flat, varying between 30 and 40 m (bLAT), and does not display a preferential slope direction (Figs. 3 and 8C), therefore restricting the ability for ice to flow quickly. The presence of dead ice also likely hindered development of a clear

drainage network and favoured the formation of pools and small deglacial lakes that were progressively filled by fine outwash sediments as the ice melted. Beets and Beets (2003) analysed the deposits preserved in a deglaciation lake in the onshore Amsterdam Basin, which include materials produced during the early phases of deglaciation. They proposed sustained presence of extensive dead-ice fields during the late Saalian to explain the severe winters recorded in lake varves. Rappol et al. (1989) also suggested that the (Saalian) ice sheet in the North Sea to have become stagnated once ice streaming in the

Norwegian Channel switched on (Sejrup et al., 2003) and disconnected the southwest sector of the ice sheet (i.e. the southern North Sea) from its source. Although the detailed dynamics of ice-sheet retreat are unknown, ice stagnation in this area seems to have been extensive, suggesting the ice progressively thinned and disintegrated at this location.

## 5.3 The post Saalian landscape

The Saalian glaciation episode in the southern North Sea creates an inherited topography that influenced subsequent landscape

evolution (Fig. 9C). Onshore, the Last Interglacial (Eemian) Rhine River invaded the IJssel glacial basin and advanced further to the north as the ice sheet retreated, finally draining into the modern North Sea approximately northwest of HKN (Fig. 9C,

Busschers et al., 2008, 2007; Peeters et al., 2016). Some of the tongue-shaped basins onshore form isolated depressions that turned into large lakes, and subsequently brackish lagoons as the sea progressively inundated the terrestrial landscape (Beets and Beets, 2003; Beets et al., 2006; van Leeuwen et al., 2000). The influence of the antecedent topography is observed in our study area by the spatial distribution of U3, reflecting the depth and relief of S1 and the glacial-fluvial deposits (U2). U3 is thicker to the south, where S1+S2 are deeper, displaying a complex internal structure and high variability of facies. However, it thins northward where glaciofluvial deposits and glaciotectonic ridges are preserved at shallow depths beneath the seafloor (Fig. 2). The influence of the antecedent topography is also reflected at a finer scale; for example, U3 in the northern sector of HKN is thicker where it directly overlies the deeper sections of the subglacial meltwater channels (Fig. 4). Future studies of the Eemian and younger sediments (which in this paper are grouped into U3 for simplicity) must consider the underlying Saalian landscape.

### 5.4 Implications of using windfarm data for palaeogeographic reconstructions

The high-density grid of seismic reflection data available from these new offshore windfarms permits detailed 3D mapping and investigation of preserved palaeo-landscapes. Careful evaluation of new datasets collected in HKN and HKZ reveals a buried glacial landscape that allows us to identify the limit of the maximum Saalian ice advance in this region ~50 km north of where it was previously inferred (Fig. 8B). Large ice-pushed ridges were systematically included in previous palaeogeographic reconstructions, but the new data revealed the absence of evidence to support such interpretation, highlighting the need to revise palaeogeographic reconstructions using legacy seismic reflection data. As new high-quality data from windfarms in the North Sea is released and evaluated, large areas are being reinterpreted, which greatly improve our understanding of their geological evolution (e.g. Cotterill et al., 2017; Eaton et al., 2020; Emery et al., 2019; Mellett et al., 2020). The complexity of the near-surface geology of the North Sea is becoming evident. Major changes in the sediment properties occur at meter scale and caution should be taken when using legacy seismic reflection data to evaluate potential offshore windfarm sites.

Our new reconstruction of the late Saalian (MIS 6) glaciation ice limit places it closer to the inferred Elsterian (MIS 12 and/or 10) ice-sheet limit in this sector of the southeastern North Sea Basin (Fig. 1). The new high-resolution data acquired to support windfarm development also shows a new level of detail on the seismic and lithological characteristics of the near-surface
geology, which permits better characterisation of late Quaternary glacial and interglacial sediments, allowing us to distinguish between the Elsterian and Saalian glaciations. This detail will allow development of a more robust stratigraphic framework enabling basin-wide correlation with areas that have a strong chronological control. High-resolution offshore windfarm data

has permitted us to better understand the distribution of the late Saalian glaciogenic and glaciofluvial deposits, which is critical to understanding the transgression of the North Sea Basin during the late Saalian/early Eemian, the influence on the distribution of depositional environments, and the preservation potential of palaeogeographical and palaeoclimatological archives.

## 6. Conclusions

Using a dense grid of high-resolution seismic reflection data acquired to support the development of offshore windfarms, we

present sedimentological, seismic stratigraphic and geomorphic analyses of a preserved glacial landscape buried in late Quaternary sediments, offshore the Netherlands.  The data record a major glaciogenic unconformity generated by ice-driven erosion during the later Drenthe glaciation of the Saalian glacial stage (mid MIS 6). Ice-sheet advance and retreat eroded and deformed the underlying strata, which resulted in a complex ice-marginal landform assemblage. A set of NE-SW oriented subglacial channels indicate that the ice likely advanced into the study area from the NE during the main advance of Saalian

land ice in this region. Till deposits are widespread in and around the subglacial channels. We interpret the large E-W composite ridge of glaciotectonised strata observed in the middle sector of the study area is as a terminal glacial moraine based on its structural style and preservation and the lack of evidence for ice advance any further south, refining the ice limit ~50 km north of its previously mapped position. Our mapping identifies a lobate shape to the ice margin, which is a consequence of contrasting subglacial bed conditions.

Following ice retreat, subglacial channels were exposed and turned into pools and small lakes, trapping fine-grained outwash sediments (Schaarsbergen Member of the Drenthe Fm.), with a diffuse drainage network of outwash channels identified from seismic mapping. The net direction of outwash is to the west or the southwest, towards a glacial basin to the west. We suggest that the preserved landscape assemblage is indicative of ice margin stagnation above a low relief landscape, probably regionally extensive in the southern North Sea. Subglacial topography may have exerted a strong control during the deglaciation phase –

explaining the stagnant remnant ice in our offshore study area and the nearby onshore (tongue basins of the Netherlands),

Earth **Surface**
**Dynamics**
Discussions

contemporary with perceived ice streaming further north (central North Sea) and northeast (Hondsrug flutes and Drenthe-II readvance margins in German Bight). The relatively horizontal and flat subglacial topography in the study area may have favoured the development of a stagnant ice margin where the ice progressively disintegrated and melted. This reconstructed MIS 6 ice marginal extent is a valuable addition to the reconstruction of the European ice sheet during MIS 6.

**Data availability**

Geophysical surveys of the HKN and HKZ windfarm development areas were carried out by Fugro between 2016 and 2018 on behalf of The Netherlands Enterprise Agency (RVO). The datasets generated during these surveys are publicly available and stored by RVO (www.offshorewind.rvo). These datasets are licensed under Creative Commons 4.0 CC-BY-SA.

**Author contribution**

VC undertook the research and wrote the article. NLMB, DMH, FSB, KMC, BLMM and WPK provided input on the article and supported the original research.

**Competing interests**

The authors declare that they have no conflict of interest.

**Funding**

This project has received funding from the European Research Council (ERC) under the European Union's Horizon 2020 research and innovation programme (grant agreement No 802281).

**Acknowledgements**

This paper forms a contribution to the ERC-funded RISeR project. We thank the Netherlands Enterprise Agency (RVO) for providing the data used in this study free of charge, licensed under Creative Commons 4.0 CC-BY-SA. We also like to thank

Keith Richards (KrA Stratigraphic) for fruitful discussions regarding the pollen data. The authors acknowledge PALSEA, a





working group of the International Union for Quaternary Sciences (INQUA) and Past Global Changes (PAGES), which in turn received support from the Swiss Academy of Sciences and the Chinese Academy of Sciences.

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
