# Peer review of "Sedimentary architecture and landforms of the Late Saalian (MIS 6) ice sheet margin, offshore The Netherlands"

_Earth Surface Dynamics, 2021_

## Referee Comment (RC1)

Cartelle et al., Sedimentary architecture and landforms of the Late Saalian (MIS 6) ice sheet margin, offshore the Netherlands

**General comments**

The paper is well written and presents timely, interesting and new insights into the lateral extent and ice marginal dynamics of the MIS 6 palaeo-ice sheet in the southern North Sea using both geophysical and sedimentological data. The paper is thorough in its assessments of available sedimentary and geotechnical data from engineering boreholes and shallow, high resolution 2D seismic and I would recommend it for publication with some relatively minor revisions listed below.

Framing of the research question or gap in knowledge that this paper will address has to be tightened in the Introduction section. At the moment when reading through the intro it sounds like the paper will tackle MIS6 ice sheet forcing mechanisms, ice sheet dynamics, spatial and temporal advance and retreat of the ice sheet and palaeo-sea-level. Of course it is fine to mention wider implications of the research but actually this paper focuses on the southerly extent of the ice sheet and ice marginal dynamics within one marine sector of the MIS6 ice sheet. This is by no means a weakness of the paper, as is eloquently described in the final section of the paper (5.4), focusing in on specific ice sheet sectors where detailed, high resolution data is available can lead to i) far greater certainty in ice sheet extent, ii) revision of previous reconstructions based on older low res data sets, iii) better interpretation/reinterpretation of ice marginal dynamics and iv) highlight the complexity of near-surface geology in the North Sea. I would try and frame the intro more towards these very important focused questions which are dealt with throughout the rest of the paper rather than far more general possible ice sheet dynamics/sea level implications.

**Specific comments**

L59: Avoid using the term "late Quaternary" throughout the manuscript. Use either Late Pleistocene or Late Pleistocene and Holocene since these are officially recognised epochs and subepochs.

L61: What time period is covered by the "late Saalian"? State this at first use. In addition state what time period or age ranges are covered by "pre-late-Saalian" and "post-Saalian".

L64-65: This relates to the comment above. Firstly how does this study investigate ice sheet morphology? This study is focused on the margin of one sector of the very large MIS6 ice sheet so I don't think you can make this claim. What do you mean by the "style" of an ice sheet as oppose to the "dynamics" of an ice sheet? I think this sentence needs a slight refocus. I don't think this paper gives so much information of broad scale advance and retreat of the ice sheet since the dataset used is from a very localised ice marginal area. Again, I don't necessarily think this is a weakness, looking at modern polar ice sheets, it is the highly complex 3-5 km marginal zone of these ice sheets that models struggle with and that are so crucial to understand ice sheet dynamics. However, I think large scale MIS6 ice sheet dynamics are beyond the remit of this paper. Thus, I would focus on the fact that you have a very detailed dataset from the ice sheet margin to allow corresponding detailed interpretation of ice margin dynamics which in turn is useful to e.g. palaeo-ice sheet modellers etc.

L93: by "glaciotectonic deformations" do you mean glaciotectonic structures?

L215: I think it is significant that mapped tunnel valleys are V-shaped, in the North Sea they can be both V- and U-shaped relating to various hydrological and/or geological factors, there are several papers on tunnel valley morphology from the North Sea (e.g. Kristensen et al. 2007; Lohrberg et al. 2020) so possibly one additional sentence could be added regarding the significance of the V-shape?

L217: why would tunnel valleys "typically form during phases of ice advance"? Surely there would be larger volumes of meltwater during deglaciation?

L218-219: What kind of "glaciotectonic deformation"? Is this a thrusted sequence? If so considering the gridded seismic data it should be possible to provide some structural measurements or the orientation of the thrusts to provide useful information on ice flow direction as has been done in the paper that is referenced by Phillips et al.

L220: Maybe worth quickly mentioning something about the peat layers from the borehole log in Fig 4 found in U1.

L260: Something that needs to be clearly explained here and throughout the manuscript is the orientation of the thrusts within the glaciotectonic ridge and the relation of these thrusts to the orientation of the ridge. From Figure 5 it seems that the thrusts within the ridge suggest ice flow from the NE while the orientation of the ridge suggests ice flow from the N.

L283: I think you have an interesting result here: a transparent or semi-transparent acoustic facies (sf1) that corresponds to a laminated and interbedded sedimentary facies. Similar transparent or sometimes termed "chaotic" acoustic facies are commonly found in the North Sea (and many other formally glaciated continental margins) and often they are simply linked to "till" or diamictons. This result highlights how important it is to be able to ground truth seismic interpretations. This very point, that transparent or chaotic acoustic facies cannot be assumed to represent till was made in a paper by Stewart and Stoker (1990, *Problems Associated with Seismic Facies Analysis of Diamicton-Dominated, Shelf Glacigenic Sequences*) where they demonstrated that several stratified glaciomarine sequences can give a similar transparent/chaotic acoustic response. Here it seems you have evidence of this and even more diverse sets of laminated/interbedded sediments that can appear acoustically transparent. It may be worth an extra sentence highlighting this here or in the interpretation.

L301: what is "slightly erosive"?

L356: Here it is stated that chronostratigraphic connection was made between the study site and onshore chronological frameworks but earlier in the manuscript it is stated that "The low resolution sub-sampling for palynological assessment in the RVO-commissioned surveys prevented a detailed correlation with established chrono-biostratigraphic frameworks." This seems somewhat contradictory.

L340-349: I wonder if there is any explanation for the lack of till in U1? Obviously S1 is a product of glacial erosion but it is interesting that only a thin till layer is present in borehole HKN10. Was this the case for other boreholes in HKN? Acoustic facies sf4 corresponds to till but that seems to be laterally discontinuous? Another point, sf4 appears to be relatively uniform in thickness at least from the seismic profiles? All of this would have interesting wider implications for subglacial processes/ice flow (i.e. the "bed mosaic" model of ice flow and depth of deformation).

L448: ridge morphology suggests ice flow from N, and thrusts from the NE, see comment above.

L478-479: but detailed structural analysis of the thrust blocks within the ice push ridge may indicate variation in flow direction?

L507-512 and figure caption 9: There seems to be quite a bit of speculation here regarding ice sheet margin stagnation. Just because there is evidence of dead ice does not necessarily mean stagnation of the ice sheet margin. Surely a rapidly retreating ice margin could also leave behind areas of dead ice. Secondly, while the NCIS switched on and off numerous times during multiple ice advances throughout the Mid and Late Pleistocene, it is not known when streaming switched on or off during individual ice advances and what influence the NCIS had on other parts of the ice sheet in terms of ice dynamics. The referenced Sejrup et al. (2003) paper relates to specific dynamics of the NCIS. For interpretations of ice margin retreat style in this SW sector of the ice sheet, either better geomorphological data or chronological data is needed.

L527: Section 5.4 is strong, I think the questions relating to these finding should be more clearly integrated with the aims in the Introduction (see general comments above).

L563: again I assume "ice margin stagnation" is linked to dead ice but could surely equally relate to rapid retreat?

L567-568: why would "flat subglacial topography" equate to stagnation of an ice sheet margin? There are numerous examples of rapid ice retreat across "flat" continental shelf areas. Several areas of the North Sea, Irish Sea, Norwegian Sea, Barents Sea etc. were deglaciated relatively rapidly after the LGM with a highly dynamic ice margin over the continental shelf. In fact I am not aware of any direct geological or chronological evidence that suggests ice margin stagnation during retreat along the NW European continental shelf other than to stabilise long enough to form grounding zone wedges or ice marginal moraines.

**Comments relating to Figures**

Fig 1. The key/legend should have a space between "Hijma et al. (2012)" and "Batchelor …" The colour schemes in several of the figures don't seem to correspond very well. In fig 1 till is outlined in green, in fig 2 till is a stippled yellow/brown and diamicton interpreted as till in the core log HKN10 is dark grey. Why not stick to dark grey for diamicton/till deposits for all figures.

Fig. 3 Again, ice-pushed ridges appear purple in my version whereas the same feature in other figures is pink.

Fig 4. Both clay and U3 appear to be the same green colour, they should be different colours.

Fig 5. Is there any diamicton in core log HKN56? If not it should be removed from the key. In the figure caption should it read "In seismic section C–C', the side of the ridge…."?

Fig 6. Again, diamicton appears to be obsolete in the key so remove.

Fig 8A. Again, colours are slightly confusing, the same glaciotectonic ridge is light grey in Fig 5 but here it is dark grey. Choose one and be consistent. Also, since you provide a colour key for all other units also include a legend for U1, even if it just says multiple sedimentary facies (I realise it is more challenging to group these sediments).

Fig 8B and 9B. At first glance the reader may wonder where the ice-push ridges across HKZ shown in Fig 1 are. Obviously you have reinterpreted these features as glaciofluvial so I think you should include

them here in Fig 8B and 9B in a different colour and label them as such to highlight that this too is a key finding of your study, i.e. reinterpretation of legacy data.

**Technical corrections**

L59: Delete space after "investigations".

L89: remove space after "Offshore".

L122: "and" should be "at"? and "conducted" should be "drilled".

L371: replace "or" with "of".

L475: should be "..observed to the northeast..".

L556: remove "is"

---

## Referee Comment (RC2)

**Sedimentary architecture and landforms of the Late Saalian (MIS 6) ice sheet margin, offshore t Netherlands**

Víctor Cartelle1, Natasha L.M. Barlow1, David M. Hodgson1, Freek S. Busschers2, Kim M. Cohen3, Bart M.L. Meijninger2, Wessel P. van Kesteren4

[referee-annotated manuscript omitted]

---

## Author Comment (AC1)

Response to Anonymous Referee #1

**General comments**

The paper is well written and presents timely, interesting and new insights into the lateral extent and ice marginal dynamics of the MIS 6 palaeo-ice sheet in the southern North Sea using both geophysical and sedimentological data. The paper is thorough in its assessments of available sedimentary and geotechnical data from engineering boreholes and shallow, high resolution 2D seismic and I would recommend it for publication with some relatively minor revisions listed below.

Framing of the research question or gap in knowledge that this paper will address has to be tightened in the Introduction section. At the moment when reading through the intro it sounds like the paper will tackle MIS6 ice sheet forcing mechanisms, ice sheet dynamics, spatial and temporal advance and retreat of the ice sheet and palaeo-sea-level. Of course it is fine to mention wider implications of the research but actually this paper focuses on the southerly extent of the ice sheet and ice marginal dynamics within one marine sector of the MIS6 ice sheet. This is by no means a weakness of the paper, as is eloquently described in the final section of the paper (5.4), focusing in on specific ice sheet sectors where detailed, high resolution data is available can lead to i) far greater certainty in ice sheet extent, ii) revision of previous reconstructions based on older low res data sets, iii) better interpretation/reinterpretation of ice marginal dynamics and iv) highlight the complexity of nearsurface geology in the North Sea. I would try and frame the intro more towards these very important focused questions which are dealt with throughout the rest of the paper rather than far more general possible ice sheet dynamics/sea level implications.

We thank the reviewer for their clear, insightful, and constructive review comments. We have replied individually to specific remarks and technical corrections below. We have also included the line numbers corresponding to the tracked changes version of the revised manuscript.

We have rewritten the first paragraphs of the introduction in response to RC1's remarks on our framing, also following suggestion from another reviewer. In the new version we introduce early on that the focus of the paper is to study a distal sector of the ice sheet where data of high resolution is available and highlighted in the framing the relevance of preserved landforms and deposits to reconstruct ice-sheet extent and dynamics. We have also clarified the objectives of the paper, now more specific and including findings from section 5.4.

**Specific comments**

L59: Avoid using the term "late Quaternary" throughout the manuscript. Use either Late Pleistocene or Late Pleistocene and Holocene since these are officially recognised epochs and subepochs.

Thanks for the advice, we have changed the terms accordingly. (Lines 68, 96, 601 and 610)

L61: What time period is covered by the "late Saalian"? State this at first use. In addition state what time period or age ranges are covered by "pre-late-Saalian" and "post-Saalian".

We have included that the late Saalian corresponds to MIS 6 the first time that the late Saalian is mentioned. (Line 71).

L64-65: This relates to the comment above. Firstly how does this study investigate ice sheet morphology? This study is focused on the margin of one sector of the very large MIS6 ice sheet so I don't think you can make this claim. What do you mean by the "style" of an ice sheet as

oppose to the "dynamics" of an ice sheet? I think this sentence needs a slight refocus. I don't think this paper gives so much information of broad scale advance and retreat of the ice sheet since the dataset used is from a very localised ice marginal area. Again, I don't necessarily think this is a weakness, looking at modern polar ice sheets, it is the highly complex 3-5 km marginal zone of these ice sheets that models struggle with and that are so crucial to understand ice sheet dynamics. However, I think large scale MIS6 ice sheet dynamics are beyond the remit of this paper. Thus, I would focus on the fact that you have a very detailed dataset from the ice sheet margin to allow corresponding detailed interpretation of ice margin dynamics which in turn is useful to e.g. palaeo-ice sheet modellers etc.

We have removed the first paragraph of the introduction, which was too broad, and instead introduce early that the focus of the paper is the study of a distal sector of the MIS 6 ice sheet. We have also highlighted the importance of preserved landforms and deposits to reconstruct past ice sheet extent and dynamics, and the relevance of investigating specific sectors where high-resolution data is available. We have also clarified the objectives of the paper, and are now more specific: "We use these data to revise previous reconstructions based on older or low-resolution datasets, to constrain the maximum extent of the ice sheet in the marine sector, to provide insights into the regional marginal dynamics of the ice sheet and to investigate the complexity of the near-surface geology of the North Sea and its implications for offshore infrastructure development and palaeogeographical reconstructions." (Lines 73-77).

L93: by "glaciotectonic deformations" do you mean glaciotectonic structures?

Yes, we have changed it accordingly. (Line 106)

L215: I think it is significant that mapped tunnel valleys are V-shaped, in the North Sea they can be both V- and U-shaped relating to various hydrological and/or geological factors, there are several papers on tunnel valley morphology from the North Sea (e.g. Kristensen et al. 2007; Lohrberg et al. 2020) so possibly one additional sentence could be added regarding the significance of the V-shape?

Thank you for the suggestion. We have included the interpretation of the V-shape in the discussion: "Several authors suggested that V-shaped subglacial channels are eroded mainly by pressurised subglacial meltwater rather than direct glacial abrasion (Jørgensen and Sandersen, 2006; van der Vegt et al., 2012)." (Lines 492-493).

L217: why would tunnel valleys "typically form during phases of ice advance"? Surely there would be larger volumes of meltwater during deglaciation?

Yes, that was incorrect. We have changed the sentence, which now reads as: ", which typically form as the result of erosional processes occurring beneath continental ice sheets." (Line 234).

L218-219: What kind of "glaciotectonic deformation"? Is this a thrusted sequence? If so considering the gridded seismic data it should be possible to provide some structural measurements or the orientation of the thrusts to provide useful information on ice flow direction as has been done in the paper that is referenced by Phillips et al.

We apologise, as this is a mistake due to a misplaced reference. Deformation found in the northern sector of HKN only corresponds to distorted reflectors, slightly wavy sometimes, found in U1 (as part of seismic facies sf2). There is no thrusting in this sector, only some faulting, although difficult to characterise due to the weak acoustic signal. We have clarified the description: "The disrupted aspect of reflectors from the underlying seismic unit (U1, sf2), with some inferred faulting and possibly folding, is interpreted as glaciotectonic deformation of the pre-existing deposits". (Lines 235-236).

L220: Maybe worth quickly mentioning something about the peat layers from the borehole log in Fig 4 found in U1.

The peat layers from the borehole log in figure 4 found in U1 are mentioned above in section 4.1.1. (Lines 198-200)

L260: Something that needs to be clearly explained here and throughout the manuscript is the orientation of the thrusts within the glaciotectonic ridge and the relation of these thrusts to the orientation of the ridge. From Figure 5 it seems that the thrusts within the ridge suggest ice flow from the NE while the orientation of the ridge suggests ice flow from the N.

The orientation of the thrusts and the ridge are explained in the description section above (4.2.1, lines 252-256). We have further clarified that the thrusts indicate a S to SW sense of displacement (ice flow coming from the N-NE). The orientation of the mapped ridge in figure 3 is not only consequence of the ice flow, but also the subsequent erosive and depositional processes. The northern side of the ridge has been intensively eroded by deglacial meltwater channels (Fig. 5), removing large sectors of it, and most of the western sector of the ridge only corresponds to a few preserved deposits, as the area suffered intensive erosion during deglaciation and transgression.

L283: I think you have an interesting result here: a transparent or semi-transparent acoustic facies (sf1) that corresponds to a laminated and interbedded sedimentary facies. Similar transparent or sometimes termed "chaotic" acoustic facies are commonly found in the North Sea (and many other formally glaciated continental margins) and often they are simply linked to "till" or diamictons. This result highlights how important it is to be able to ground truth seismic interpretations. This very point, that transparent or chaotic acoustic facies cannot be assumed to represent till was made in a paper by Stewart and Stoker (1990, Problems Associated with Seismic Facies Analysis of Diamicton-Dominated, Shelf Glacigenic Sequences) where they demonstrated that several stratified glaciomarine sequences can give a similar transparent/chaotic acoustic response. Here it seems you have evidence of this and even more diverse sets of laminated/interbedded sediments that can appear acoustically transparent. It may be worth an extra sentence highlighting this here or in the interpretation.

Thank you for the suggestion. We have included a few sentences about this observation as part of section 2.4 when we discuss the need to revise palaeogeographic reconstructions using legacy seismic reflection data: "Similarly, we have identified transparent and semi-transparent seismic facies, which are usually interpreted as subglacial diamicton on formerly glaciated continental margins (Stewart and Stoker, 1999), that correspond to diverse sets of laminated and interbedded glaciofluvial or coastal sedimentary facies (sf1 and sf3, Table 1). These results further highlight the importance of being able to ground truth seismic interpretations." (Lines 589-593)

L301: what is "slightly erosive"?

With "slightly" we were meaning of minor order, however, it is not needed as they are found within the seismic unit, so we have deleted the word slightly. (Line 323)

L356: Here it is stated that chronostratigraphic connection was made between the study site and onshore chronological frameworks but earlier in the manuscript it is stated that "The low resolution sub-sampling for palynological assessment in the RVO-commissioned surveys prevented a detailed correlation with established chrono-biostratigraphic frameworks." This seems somewhat contradictory.

Yes, it is somewhat contradictory, so we have decided to delete the earlier sentence from methodology section (Lines 152-153). It was present to indicate that the palynological data were

of too low resolution to do traditional pollen-based chronostratigraphy 'within interglacials', but this is a detail explaining limitations of the data. The later sentence is retained (lines 359-360). That sentence relates to the overall glacial-interglacial Pleistocene framework, separating individual cycles and tying a numeric age to them.

L340-349: I wonder if there is any explanation for the lack of till in U1? Obviously S1 is a product of glacial erosion but it is interesting that only a thin till layer is present in borehole HKN10. Was this the case for other boreholes in HKN? Acoustic facies sf4 corresponds to till but that seems to be laterally discontinuous? Another point, sf4 appears to be relatively uniform in thickness at least from the seismic profiles? All of this would have interesting wider implications for subglacial processes/ice flow (i.e. the "bed mosaic" model of ice flow and depth of deformation).

The diamicton layers were recovered in several boreholes in the northern sector of HKN, not only in HKN10, but always thin layers associated with S1. We mapped the preserved diamicton/till following seismic facies sf4 that were associated with surface S1, appearing as patchy deposits, picking S1 always at the base of such deposits. Of course, there is a degree of uncertainty in the interpretation considering that more detailed analyses and testing were not possible in the boreholes. The mapped distribution may also be strongly related to preservation and depends on the spatial resolution of the seismic grid. Given the uncertainty and limitations, we consider that it is difficult to discuss wider implications without additional data. We did not modify the text on this point. (Lines 364-372).

L448: ridge morphology suggests ice flow from N, and thrusts from the NE, see comment above.

The orientation of the mapped ridge in Figure 3 is not only a consequence of the ice flow, but also the subsequent erosive and depositional processes, and therefore, we consider that this cannot be taken as strong evidence of ice flow coming from the N. The thrust indicate ice coming from N to NE, and subglacial meltwater channels, which are larger and therefore easier to characterise in the seismic grid, indicate ice flow from NE. We did not modify the text on this point. (Lines 489-490).

L478-479: but detailed structural analysis of the thrust blocks within the ice push ridge may indicate variation in flow direction?

The thrusts blocks indicate a S to SW sense of displacement (ice flow coming from the N-NE), but it is not possible to refine this orientation even more (and distinguish N from NE ice flow) as it is limited by the spatial resolution of the seismic grid and the orientation of the seismic profiles. We have only slightly modified the text on this point following suggestions from another reviewer: "In the HKN windfarm area, there is no direct evidence for a change in ice-flow direction" (Line 525).

L507-512 and figure caption 9: There seems to be quite a bit of speculation here regarding ice sheet margin stagnation. Just because there is evidence of dead ice does not necessarily mean stagnation of the ice sheet margin. Surely a rapidly retreating ice margin could also leave behind areas of dead ice. Secondly, while the NCIS switched on and off numerous times during multiple ice advances throughout the Mid and Late Pleistocene, it is not known when streaming switched on or off during individual ice advances and what influence the NCIS had on other parts of the ice sheet in terms of ice dynamics. The referenced Sejrup et al. (2003) paper relates to specific dynamics of the NCIS. For interpretations of ice margin retreat style in this SW sector of the ice sheet, either better geomorphological data or chronological data is needed.

We have removed the section discussing ice margin stagnation and clarified the interpretation of dead ice presence in the study area, which we suggest as an explanation for the unusual deglacial landscape preserved in HKN (Lines 547-563). We have also included a new paragraph

discussing the possibility of surge-type behaviour in this marginal sector of the ice sheet in line with comments from another reviewer. (Lines 424-435 and 563-568).

L527: Section 5.4 is strong, I think the questions relating to these finding should be more clearly integrated with the aims in the Introduction (see general comments above).

Thank you for the kind words. We have integrated this a part of the rewritten objectives: "We use these data to revise previous reconstructions based on older or low-resolution datasets, to constrain the maximum extent of the ice sheet in the marine sector, to provide insights into the regional marginal dynamics of the ice sheet and to investigate the complexity of the near-surface geology of the North Sea and its implications for offshore infrastructure development and palaeogeographical reconstructions." (Lines 73-77).

L563: again I assume "ice margin stagnation" is linked to dead ice but could surely equally relate to rapid retreat?

Yes, it was linked to the presence of dead ice, however, we have now removed the discussion of ice stagnation and included a new discussion of surging glacial landsystems in line with comments from another reviewer. (Lines 424-435 and 547-568).

L567-568: why would "flat subglacial topography" equate to stagnation of an ice sheet margin? There are numerous examples of rapid ice retreat across "flat" continental shelf areas. Several areas of the North Sea, Irish Sea, Norwegian Sea, Barents Sea etc. were deglaciated relatively rapidly after the LGM with a highly dynamic ice margin over the continental shelf. In fact I am not aware of any direct geological or chronological evidence that suggests ice margin stagnation during retreat along the NW European continental shelf other than to stabilise long enough to form grounding zone wedges or ice marginal moraines.

As indicated above, we have removed that section from the manuscript. (Lines 424-435 and 547-568).

**Comments relating to Figures**

Fig 1. The key/legend should have a space between "Hijma et al. (2012)" and "Batchelor …" The colour schemes in several of the figures don't seem to correspond very well. In fig 1 till is outlined in green, in fig 2 till is a stippled yellow/brown and diamicton interpreted as till in the core log HKN10 is dark grey. Why not stick to dark grey for diamicton/till deposits for all figures.

Correction to the legend done. We apologise for the changes in the colour schemes. Till/diamicton is now dark grey in all the figures (colour changed in figures 1, 3 and 8) and glaciotectonic ridges light red (changes done in figures 3 and 8).

Fig. 3 Again, ice-pushed ridges appear purple in my version whereas the same feature in other figures is pink.

The colour of the glaciotectonic ridges has been changed to light red (same as figure 1).

Fig 4. Both clay and U3 appear to be the same green colour, they should be different colours.

We have changed the colour for clay in figures 4, 5, 6 and 7.

Fig 5. Is there any diamicton in core log HKN56? If not it should be removed from the key. In the figure caption should it read "In seismic section C-C', the side of the ridge…."?

No, there is no diamicton in HKN56 and has been removed from the key. The caption should read "side of the ridge", so the correction has been done.

Fig 6. Again, diamicton appears to be obsolete in the key so remove.

Diamicton removed from the key.

Fig 8A. Again, colours are slightly confusing, the same glaciotectonic ridge is light grey in Fig 5 but here it is dark grey. Choose one and be consistent. Also, since you provide a colour key for all other units also include a legend for U1, even if it just says multiple sedimentary facies (I realise it is more challenging to group these sediments).

We have used the darker grey colour to be able to differentiate the deformed strata from other deposits of U1. In the new version we have included the legend for U1, as older deposits with multiple facies, and we have changed the colour of the deformed strata to light red (same as glaciotectonic ridges).

Fig 8B and 9B. At first glance the reader may wonder where the ice-push ridges across HKZ shown in Fig 1 are. Obviously you have reinterpreted these features as glaciofluvial so I think you should include them here in Fig 8B and 9B in a different colour and label them as such to highlight that this too is a key finding of your study, i.e. reinterpretation of legacy data.

We have added the channel and bars fills identified in HKZ to figure 8B, to highlight our reinterpretation of these deposits. However, we have not included these bars in figure 9B as most of these are found as part of unit U1 and therefore correspond to a time before that depicted in figure 9B.

**Technical corrections**

L59: Delete space after "investigations".

Space deleted. (Line 68)

L89: remove space after "Offshore".

Space deleted. (Line 103)

L122: "and" should be "at"? and "conducted" should be "drilled".

Changes done. (Line 135)

L371: replace "or" with "of".

It should be "or" as it refers to two different terms usually found in the literature for the same feature. (Line 394)

L475: should be "..observed to the northeast.."

Changed to "..observed in the northeast…". (Line 521)

L556: remove "is"

Removed. (Line 616)

---

## Author Comment (AC2)

Response to David Evans

**General comments**

This is an important report on, and interpretation of, new offshore data relevant to the reconstruction of MIS 6 glaciation of the North Sea. The interpretations are largely entirely logical and valid but I feel that the authors do not quite make the most of this important database in terms of implications for former glacier dynamics. A significant issue pertaining to the glaciation styles in the North Sea is the potential of surging versus normal active ice recession - this has been debated often for the MIS 2 glaciation and has been re-invigorated by the increasing amounts of offshore data becoming available. Particularly pertinent are the occurrences of large glaciotectonic thrust masses and widespread ice stagnation evidence (the latter not as convincingly demonstrated as the former in this paper), both diagnostic of surging glacial landsystems. The authors opt for active recession but the evidence begs further evaluation in this respect. In terms of details on glaciotectonic landforms, the authors might also want to consult some recent literature on the development of hill-hole pairs, whereby thrust masses aligned parallel to basins are termed paraxial ridges and are part of the overall development of hill-hole pair development and hence are not mutually exclusive with ridges aligned perpendicular.

I have added a number of queries and corrections on the pdf of the manuscript, which I attach here.

We thank the reviewer for their detailed, constructive, and useful comments. We have addressed all of the specific comments below and we have included the line numbers of the tracked changes version of the revised manuscript.

We have added a paragraph discussing surging as the origin of the thrust masses found in our study area. We have also clarified our interpretation of the presence of dead ice in the area and related it to the possibility of a surging ice-sheet terminus. We have also included references to supporting literature for such interpretations, and new references about the growing evidence of surge-type behaviour for ice-sheet masses in the North Sea. Additionally, we have also included the "paraxial ridges" terminology in the discussion and the appropriate references.

**Specific comments**

Title: Here and throughout the text make sure that this is capitalised = The Netherlands. It is incorrect in many places.

Thank you for the correction. We have changed it throughout the text. (Lines 12, 19, 56, 99, 114, 398, 485, 521, 611).

L24-25: This is an ugly sentence. Try: .....data as records of process-form relationships preserved in buried landscapes, which can be utilised in refining palaeo-ice sheet margins and informing longer term drivers of change in low relief settings.

Thank you for the suggestion, we have changed the sentence. (Lines 24-25).

L27-37: This opening paragraph really doesn't work and because it is a token gesture on applications to future climate change predictions it is entirely out of context and provides no lead into what follows. The logic is also confused, because we use modern analogues to reconstruct the past, not the reverse - even if we did use ancient examples we would not use such a fragmentary record as that of MIS6! In short, you can pretty much just delete this paragraph.

We have deleted the paragraph and unused references. (Lines 29-39).

L52: So is yours! I'd be careful in what you imply here.

We imply that even onshore studying small sectors is sometimes the only option. Adding those up is what finally allows to get more robust and precise reconstructions. We have slightly changed the text regarding that point to make it clearer: "Over Germany and The Netherlands, the MIS 6 maximum ice advance was more extensive than the LGM and consequently relatively well preserved; and is known as the Saalian Drenthe substage ice limit in the regional stratigraphic schemes. These ice limits have been studied from spatially disparate field evidence onshore over a series of ice sheet subsectors". (Lines 56-60).

L84-85: add Dove et al 2017 here, as it pertains to your point and uses similar data types.

Reference added. (Line 98).

L206: What is a concave-up depression? This sounds like a contradiction in terms. Do you mean depression with concave-up floors?

This is probably confusing. We have simplified it, removing "concave-up". Now it reads: "above isolated depressions". (Line 222)

L217: A rather restricted and non-primary set of references on such a large topic. At least cite O' Cofaigh 1996 and maybe also Wingfield 1996 and Clayton et al 1999

We have added the following references which describe features of similar dimensions to those found in our study area: Clayton et al., 1999 and Ó Cofaigh, 1996. (Line 233).

L219: This is not specific enough - it is not actually loading, as that implies normal stress only. These features are created by compressive folding and thrust stacking. Phillips et al 2018 is a good example but you need to include some primary concept citations here also - like Mulugeta & Koyi 1987, Aber et al 1989, van der Wateren 1995

We apologise, as this is a mistake due to a misplaced reference. Deformation found in the northern sector of HKN only corresponds to distorted reflectors, slightly wavy sometimes, found in U1 (as part of seismic facies sf2). There is no thrusting in this sector, only some faulting, although difficult to characterise due to the weak acoustic signal. Thus, we have removed the reference to Phillips et al. (2018) and clarified the description and interpretation as follows: "The disrupted aspect of reflectors from the underlying seismic unit (U1, sf2), with some inferred faulting and possibly folding, is interpreted as glaciotectonic deformation of the pre-existing deposits." (Lines 235-236).

L221: This is a risky business! You can't definitively interpret the genesis of a diamicton from seismic records! The term "glacial" when put before the term "till" is also a redundancy. I suggest that this reads: ...are interpreted as glacigenic deposits, probably subglacial traction till.

Thank you for the suggestion. We have applied the suggested change in the sentence: "The patchy high-amplitude reflectors found at the base of the subglacial meltwater channel-fills, corresponding to matrix-supported diamicton in the boreholes, are interpreted as glaciogenic deposits, probably subglacial traction till." (Lines 237-238).

L227: ...are likely to be... These can't possibly be subglacial. How would vegetation with oak trees exist up-ice in a landscape covered by an ice sheet? This makes no sense. I suggest you use: ...are likely to be channel infills laid down after deglaciation.

Yes, that was a mistake. We have modified it following the suggestion: "Given the seismic architecture, these sediments are likely to be channel infills laid down after deglaciation." (Lines 244-246).

L236: compressional deformation surely?

Yes, the ridge is consequence thrust stacking.

L237: There is no such word as "thrusted". This is a common error. The word is always "thrust" or "thrusting". This needs correcting in a number of places from hereon.

We have replaced the word "thrusted" with "thrust" throughout the manuscript. (Lines 257, 262, 280, 271, 291 and 417)

L377: is this not Ijssel?

No, the correct term is IJssel, with capital I and J. This is because in Dutch the vowel IJ is one character.

L396: This is incorrect. The ridges represent the crests of individual thrust slices or fold noses - they are not indicative of active retreat. In fact large thrust masses are almost exclusively associated with surging - this needs to be considered and discussed, as you are likely looking at a surging glacial landsystem.

We have corrected the section regarding the interpretation of glacial landforms described by Mellett et al. (2020) and Phillips et al. (2018). It now reads as: "In the Dudgeon windfarm and the eastern sector of Dogger Bank, the glaciotectonised sequences form multiple parallel ridges extending over several kilometres, which are interpreted to be the result of surge-related marginal readvances during overall ice-sheet retreat (Mellett et al., 2020; Phillips et al., 2018)." (Lines 418-420).

We have also included a new paragraph discussing the interpretation of HKN as a surging glacial ice-sheet terminus: "Large thrust-block moraines are usually found at the margins of surging glacial landsystems formed due to rapid ice advance into proglacial and pre-existent sediments (Evans and Rea, 2005,1999). Surge-type behaviour has gained relevance in the discussion of marginal dynamics of former ice sheets (e.g., Bateman et al., 2015; Boston et al., 2010; Evans et al., 2019, 2020; Graham et al., 2009; Mellett et al., 2020; Phillips et al., 2018; Vaughan-Hirsch and Phillips, 2017). In HKN, a single thrust-block moraine is preserved and likely formed during a surge event when rapid advance of the ice sheet led to the pressurisation of groundwater within the underlying Quaternary sediments. In the northern sector, large subglacial meltwater channels (Figs. 3 and 4) were eroded in response to the over-pressurisation, while in the middle sector, the thick and laterally extensive mud and peat layers facilitated the development of a décollement and thrust staking in front of the advancing ice mass (Figs. 3 and 5). Although glaciotectonic thrust moraines cannot be taken as solely diagnostic of surging activity (Evans and Rea, 2005,1999), the glacial landforms identified in HKN (Fig. 3) is compatible with surging activity and therefore potentially indicative of a scenario of ice-marginal instability triggered by internal ice sheet dynamics rather than by external climatic forcing." (Lines 424-435).

L435: These are called paraxial ridges and are not mutually exclusive - they are part of hill-hole pair evolution. See Evans et al 2021 in QR for the establishment of this term and examples.

Thank you for the suggestion. We have included the term paraxial ridges (and the reference Evans et al., 2021) to describe ridges parallel to the glacial basins and removed the sentence pointing to these features being mutually exclusive. Now it reads: "The glaciotectonic ridges in HKN windfarm are oriented perpendicular to the P/Q-block basin, whereas onshore in the central Netherlands, there are large ridges that formed parallel to the basin rims. The later

correspond to paraxial ridges which are part of the evolution of hill-hole pairs (Evans et al., 2021). The glaciotectonic ridge identified in HKN seems to continue beyond the limits of the windfarm, and geomorphological features similar to the paraxial ridges may also be present beyond the areas covered by the dataset." (Lines 472-4477).

L446: This is the first time we see the term V-shaped - surely they are U-shaped if they are subglacial/tunnel channels?

In the North Sea, tunnel valleys have been described both as U- and V-shaped in cross section using seismic data, see for example Kristensen et al. (2007), Lohrberg et al. (2020), Stewart et al. (2013), and van der Vegt et al. (2012). In the case of the features identified in HKN, they are V-shaped in cross section in the seismic profiles (Fig. 4A). We did not modify the text on this point. (Lines 489-490).

Kristensen, T. B., Huuse, M., Piotrowski, J. A. and Clausen, O. R.: A morphometric analysis of tunnel valleys in the eastern North Sea based on 3D seismic data, J. Quat. Sci., 22(8), 801–815, doi:10.1002/JQS.1123, 2007.

Lohrberg, A., Schwarzer, K., Unverricht, D., Omlin, A. and Krastel, S.: Architecture of tunnel valleys in the southeastern North Sea: new insights from high-resolution seismic imaging, J. Quat. Sci., 35(7), 892–906, doi:10.1002/JQS.3244, 2020.

Stewart, M. A., Lonergan, L. and Hampson, G.: 3D seismic analysis of buried tunnel valleys in the central North Sea: Morphology, cross-cutting generations and glacial history, Quat. Sci. Rev., 72, 1–17, doi:10.1016/j.quascirev.2013.03.016, 2013.

van der Vegt, P., Janszen, A. and Moscariello, A.: Tunnel valleys: Current knowledge and future perspectives, Geol. Soc. Spec. Publ., 368(1), 75–97, doi:10.1144/SP368.13, 2012.

L499- 502: This section of text is actually irrelevant as you don't expand on ice streaming anywhere else. Delete from "Geomorphological evidence.......  to  ....Winsborrow et al 2010)." And remove any of the references not used other than here.

We have deleted the sentence and the unused references. (Lines 547-550).

L504: Unless it surged? This concept needs to be examined. This is the first place that dead ice is mentioned - what is the evidence for this? If it is indeed strong, then you have another diagnostic criteria for surging.

We have removed the sentences about ice streaming, and a new paragraph discussing surging has been added to the previous section (Lines 424-435). We have clarified the interpretation of this section and the explanation for the presence of dead ice. This is inferred to explain the complex deglacial landscape, although other interpretations are feasible. This now reads: "Ice retreat left an unusual landscape preserved in HKN, characterised by the progressive filling of ice-advance and glacio-fluvial drainage depressions with laminated deposits recording postglacial climatic change through the preserved pollen (Figs. 3 and 4). The landscape is also characterised by a diffuse drainage network where only a few small deglacial channels are identified (Figs. 3 and 5). We interpret that this proglacial landsystem developed due to the presence of dead ice in HKN during deglaciation. Large masses of dead ice likely hindered development of a clear drainage network and favoured the formation of pools and small deglacial lakes that were progressively filled by fine outwash sediments as the ice melted. This interpretation is also coherent with a surge-type glacial behaviour, as distal parts become stagnant after a surge event leaving large masses of dead ice to melt in the formerly glaciated area. However, to improve understanding of ice margin retreat style in this southwest sector of the ice sheet, additional geomorphological and chronological data is needed, particularly towards the west and north of HKN and HKZ". (Lines 552-568).

L508: This is just illogical - how does dead ice create severe winters? Consider deleting this as it makes no sense.

This sentence has been deleted. (Lines 558-559).

L511: Again - why? You need to provide evidence for stagnation.

As indicated above, we have removed the interpretation of ice margin stagnation, as this was a little bit speculative, and included discussion if ice surging. (Lines 552-568).

L539-540: consider citing Gibbard et al. 2018 (Royal Society Open Science) and Evans et al 2019 (PGA) here on the MIS 6 limit in eastern England.

We have not included such references here, as the section is on the Scandinavian-sourced southwesterly sectors of the ice-sheet in the North Sea and The Netherlands, not the more westward British-sourced parts of the ice-sheet. Therefore, we have not modified the text. (Lines 598-599).

L569: Surging needs to be a significant element of your conclusions.

Surging has been included in the conclusions: "We suggest that the preserved landscape assemblage is indicative of a surging glacial ice-sheet terminus. The thrust-block moraine preserved in the study area likely formed during a surge event when rapid advance led to the pressurisation of groundwater within the underlying Quaternary sediments, which led to the erosion of large subglacial meltwater channels and thrust stacking in front of the advancing ice mass. Surge-type behaviour in this distal sector of the ice sheet indicate ice margin instability independent of external climate forcing." (Lines 622-628).

**Comments relating to Figures**

Figure 5: The single barbed thrust arrow symbols on the figures from hereon are too small. Enlarge to at least 3 times the size.

We have increased the size of the arrows.

Figure 8C: Glacitectonic ridge not ice pushed. Sea ice can push ridges so this term is very ambiguous.

We have changed the name to glaciotectonic ridge in figures 1, 3, 8 and 9.

Figure 9: deglacial not deglaciation, glacitectonic ridges. Delete (this study) - obviously it's your figure!

Corrections done and caption changed. (Lines 499-512).

**Technical corrections**

L19: and displaying

We have added "and". (Line 19).

L48: Delete "broadly speaking" =  The MIS 6.....

We have deleted "Broadly speaking". (Line 54).

L52: relates

We have changed this sentence and the word "relate" is no longer present. (Line 60).

L59: close up space, ...stimulus for new research into late Quaternary submerged landscapes.

Changes done. (Line 68).

L63: ...offshore of the...

We have slightly modified this sentence. (Line 73).

L64: ..these data....

We have changed "this data" to "these data". (Line 73)

L79: basin, reaching

Comma added. (Line 92).

L93: ...deformation structures are....

Change done. (Line 106).

L184: ...were recovered....

Correction done. (Line 199).

L193: Surface S1 is.....

We have removed "In the northern sector,". (Line 209).

L194: elongate

Correction applied. (Line 210).

L226: ...the top of U2,...

We have added "of". (Line 244).

L233: ...of the HKN...,  ...are faulted and distorted

Changes done. (Line 252).

L260: ...of thrust, faulted and folded....

Changes done. (Line 280).

L265: ...NE-SW drainage direction....

Word "drainage" added. (Line 285).

L270: overthrust

Changed. (Line 291).

L316: indicating deposition...

Change done. (Line 339).

L334: ...offshore of the... or "offshore The Netherlands"

Changed to "offshore of the Dutch coast". (Line 357).

L335: ice sheet-driven

Change done. (Line 358).

L370: offshore of the

Word "of" added. (Line 393).

L373: ..margins and marking...

Word "and" added. (Line 396).

L375: ...onshore glacial record in The Netherlands,....

Change done. (Line 398).

L378: stacked

Change done. (Line 401).

L405: ...comprised south....

"Of" removed from the sentence. (Line 441).

L410: comprising a thin....

Word "comprising" added. (Line 446).

L433: Glacitectonic ridge not ice pushed. Sea ice can push ridges so this term is very ambiguous.

"Ice push ridges" changed to "glaciotectonic ridges". (Line 455).

L434: Ditto

"Ice push ridges" changed to "glaciotectonic ridges". (Lines 471-472).

L445: offshore of the Dutch coast...

Change done. Now it reads: "into the nowadays offshore regions of the Dutch coast". (Line 488).

L449: ...sediments, leading....

Comma added. (Line 494).

L450: ..thrust-bound sediment packages (slices)

Word "packages" added. (Line 495).

L468: Unnecessary and ambiguous wording.  ...1989), forming marginal.....

Unnecessary words removed ("releasing englacial debris"). (Line 514).

L469: A further advance then formed the maximum....Netherlands, creating the offshore HKN.....

Thank you for the suggestion. Change done. (Line 516).

L471: and the further

Word "and" added. (Line 518).

L472: position, the dominant

Word "also" deleted. (Line 518).

L473: HKN also suggests

Word "also" added. (Line 519).

L474: readvance followed

Word "has" deleted. (Line 520).

L475: observed in the northeast

Change done, word "in" added. (Line 521).

L478: ...area, there is no direct evidence for a change in ice flow direction. The......

Change done. (Line 525).

L486: eroded

Correction done. (Line 533).

L490: which, with........level, likely.....

Commas added. (Line 537).

L506: deglacial lake

Correction done to figure 9.

L509: become stagnant

This part of the text was removed. (Line 561).

L514: ....created a topography that.......

Word "inherited" deleted. (Line 570).

L515: Ijssel?

The correct name is IJssel. No changes done. (Line 571).

L517: formed

Correction done. (Line 573).

L532: reveals

Correction done. (Line 588).

L534: improves

Correction done. (Line 594).

L543: framework,

Comma added after framework. (Line 603).

L551: ice sheet-driven

Word "sheet" added. (Line 611).

L556: the word "glacial" is redundant when put in front of moraine. Its just "terminal moraine" here.

Word "glacial" deleted. (Line 616).

---

## Author Response (AR2)

Response to Associate Editor Arjen Stroeven

**General comment**

Dear Dr. Cartelle,

I am happy to recommend publication of your manuscript in Earth Surface Dynamics pending minor editorial revisions (mostly to your figures). Please consult the annotated manuscript that I appended.

Best wishes,

Arjen Stroeven

We thank the associate editor for the detailed revision of the manuscript. We have modified the figures accordingly and made de appropriate corrections to the text. We have addressed all of the specific comments below and we have included the line numbers of the tracked changes version of the revised manuscript.

**Specific comments**

In figures: Km = km

This correction has been made in the scale bars of maps from figures 1, 3, 4, 5, 6, 7, 8 and 9.

Fig. 1: the unit is "m a.s.l.", add zero

Units (m above sea level) and zero value added to the topography scale bar.

L97: Previous research (Joon et al., 1990) suggests…

Correction made (Line 97).

Table 2: The scales are hard to read on some of these. Put an opaque or semi-transparent box behind them.

A semi-transparent white box has been added behind the scale bars.

Figure 2: explain "bpsbelow", add s to map in caption

The word "bpsbelow" was a mistake, is should read m below LAT. It has been corrected and added to the caption. The caption has also been corrected.

L177: I'm not sure where Table 2 is?

That is a mistake, it should be Table 1. It has been corrected. (Line 177).

Figure 4 caption (L186), lines = seismic profiles

We have changed "lines" for "seismic profiles" in the caption of figures 4 and 7.

L221: indicates, also following occurrences- reserve the use of "suggest" for authors, but not for evidence. Use indicate or some other verb.

Thank you for the suggestion, we have changed "suggest" for "indicate" through the text when the verb is referring to evidence (Lines 221, 325, 327, 338, 358, and 487).

Figure 5: ...in panels in other figures, you've done away with a scale in the interpretative map?

That was an error when exporting the figures to PNG. It has been corrected in figures 2, 4, and 6.

Figure 6: lamiane = laminae

Change done.

Figure 7: HKZ4-BH04

Correction done.

L363 and 560: 5. Discussion and 6. Conclusions.

Changes done.

Figure 8A: Explain the questionmark in the caption. Also why is the Sandur unit much more extensive than the outwash fan in the profile?

The extension of the sandur unit has been corrected to match that of the outwash fan. The question mark has been removed, as it is not needed.

Figure 8A: correct with sea level at -10 LAT?

Yes, we are illustrating a phase of the transgression given that the exact position of the sea-level during the highstand is under discussion. To clarify this, we have modified the caption, now it reads: "Conceptual model of the infill (following transgression from MIS 6 to MIS 5e, but before the highstand)" (Line 338).

Figure 8B: Explain the reasons for the question marks in the caption.

Question marks indicate areas of uncertainty, it has been added to the caption (Line 440).

Figure 8B: the "prime" in X' is hard to see. Move slightly lower.

Correction done.

Figure 8B: reference which of the Fig 1 interpretations of the ice margin is shown here.

References added to the caption: "(B) Comparison between the previously inferred maximum Saalian ice sheet extent (dashed line, Hijma et al, 2012, Batchelor et al., 2019)" (Line 439).

Figure 8 legends: I'm not sure the black works for glacial till? Is everything that is not "basin" or "ridge" or "channel" glacial till??

Yes, the grey/black colour is difficult to discern. We have changed the colour of the glacial till in this map (Fig. 8B) but also in the previous figures (Figs. 1 and 3), to keep it consistent.

Figure 9: Ice=ice, braid=braided-river?, Water-flow direction. why has the ridge decreased in size between panels B and C?

Changes done. The change in the size of the ridge was a drawing error, it has been corrected.

Figure 9C: (ca. xxx ka)?

We cannot add a timing for this stage as the chronology for the Eemian transgression and highstand in the North Sea is under discussion and, therefore, adding a time would require extensive discussion.